# Analyzing Few-Shot Neural Architecture Search in a Metric-Driven Framework

Timotée Ly-Manson[1]  Mathieu Léonardon[1]  Abdeldjalil Aissa El Bey[1]
Ghouthi Boukli Hacene[2]  Lukas Mauch[2]

[1]IMT Atlantique, UMR CNRS 6285 Lab-STICC, 29238 Brest, France
[2]Sony Europe B.V., Stuttgart Laboratory 1, Germany

**Abstract**  While Neural Architecture Search (NAS) methods help find optimal neural network architectures for diverse tasks, they often come with unreasonable costs. To tackle such a drawback, the one-shot NAS setting was introduced, where a supernet is used as a superposition of all architectures in the space and performs the search in a single training phase. While this method significantly reduces the cost of running NAS, the joint optimization of every architecture degrades the performance of the search. The few-shot NAS line of work tackles this issue by splitting the supernet into sub-supernets trained separately, each with a reduced level of weight-sharing, which gives rise to the new challenge of finding the best way to split the supernet. In particular, GM-NAS utilizes a gradient matching score to group operations in a splitting schema. We extend and generalize this method by building a framework with compatibility for any arbitrary architecture evaluation metric, enabling the generation of numerous and diverse splits. We leverage this new framework in conjunction with various metrics from the zero-shot NAS literature and investigate the benefits of splitting across algorithms and metrics. We find that architectures are distributed in disadvantageous ways inside splits, and that proposed supernet selection methods are flawed.

## 1 Introduction

Neural Architecture Search (NAS) [15, 19, 26], has gained traction in recent times as it holds the promise to find the optimal neural network architecture for any given task. Early methods relied on reinforcement learning [26] or evolutionary algorithms [17] and incurred unaffordable training times when scaling up. Recent works use one-shot NAS [19] to reduce the computation requirements down to a single training. In the traditional one-shot NAS setting, the search space is reduced to a single directed acyclic graph (DAG) with searchable operations on all of its edges. Training is conducted on the supernet, whose edges in the DAG constitute a superposition of all possible operations. Namely, the weights of all architectures in the space are shared for every operation they have in common.

Despite the significant improvements to search costs, one-shot NAS is criticized [2, 23] for its reduced efficacy as a proxy to candidate architecture performance and difficulties discovering better architectures in the search space. These flaws are linked to inherent properties of the joint training, namely co-adaptation. Indeed, some groups of operations may descend to undesirable directions when trained together. Few-shot NAS [8, 25] is introduced as a middle ground to reduce one-shot disadvantages while still maintaining reasonable computational costs. By partitioning the search space into smaller sets, the number of sub-supernets to train increases, but the quantity of weight-sharing in each of them is decreased. In practice, the accuracy of the sub-supernets is used as a proxy to choose a single sub-supernet to train, reducing the cost of few-shot NAS to be equivalent to that of a single one-shot NAS search.

The authors of GM-NAS [8] argue that splitting exhaustively between all operations, as proposed in [25], is wasteful as some operation groups could be less affected by co-adaptation. They propose

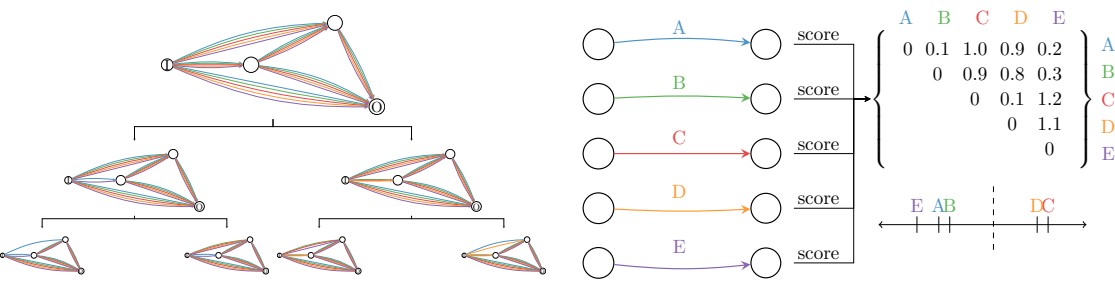

(a) Few-shot NAS splitting tree, with depth 2 and branching factor 2.
At each layer of the tree, operations on the selected edge are partitioned into 2 groups.

(b) Extended splitting procedure.
On the selected edge, using any metric, operations are individually scored and compared. Partitions are formed via min-cut optimization.

Figure 1: Full supernet splitting framework

to build sub-supernets from a splitting tree created with the help of the gradient matching score, which groups together operations whose gradients point to similar directions.

Taking inspiration from the splitting schema described in GM-NAS, we introduce a supernet splitting framework which is, to the best of our knowledge, the most general framework in the few-shot NAS field to this date. Importantly, this extended supernet splitting framework allows to use any arbitrary metric in the decision-making of the splits. We make use of metrics first seen in the zero-shot NAS [1, 3, 16] literature, which are designed with the intent of scoring architectures, and implement them as splitting metrics in our framework.

We take full advantage of the increased quantity and variety of sub-spaces and splits generated through this procedure to investigate several important assumptions for the few-shot NAS paradigm. Firstly, we verify that supernet splitting is indeed beneficial to helping one-shot NAS algorithms reach better-performing architectures. We also assess the impact of various metrics. Secondly, we observe that splitting is unable to isolate the best architectures in their own subspace, which undermines synergies with search algorithms. Finally, we show that commonly used proxies to subspace selection, such as the accuracy of the supernet, are flawed and cannot recognize a good supernet, much less indicate which contains the best architecture in the space.

Our code is available at `https://github.com/brain-bzh/metric_driven_few_shot_nas`.

## 2 Related Work

### 2.1 One-Shot NAS

Traditional NAS methods rely on reinforcement learning-based predictors [26], evolutionary algorithms [17] or Bayesian kernels [21], all of which are computationally heavy as they require the training of many candidate networks to act as labels in the search.

Early on, weight-sharing is proposed as an alternative way to conduct NAS in a single training [19], leading to the parallel research branch known as one-shot NAS. In the one-shot NAS paradigm, it is sufficient to train the supernet, which represents any candidate network in the search space. This change in representation enables the joint training of all candidate architectures with a single training phase.

Many methods build on top of the weight-sharing mechanism to discover new candidate architectures. A popular setup [4, 7] is to utilize the trained supernet as a proxy for the search. The weights of the supernet are copied onto a sampled candidate architecture to estimate its performance without training and guide the search. Other lines of work aim to discover an architecture at the same time as training. DARTS [15] equips the supernet with architecture parameters learned in a differentiable manner. While this approach is simple and effective, it fails to find highly performing

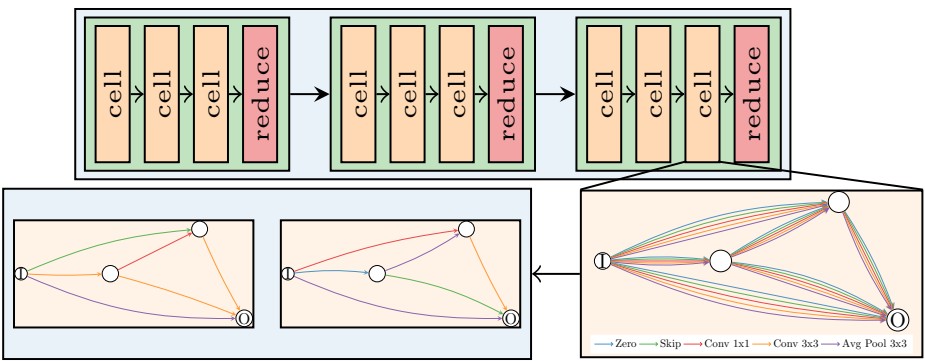

Figure 2: Structure of the supernet in the NAS-Bench-201 [6] search space. Candidate architectures are sampled from the supernet by discarding all but one operation on each edge.

architectures as it hyperfocuses on non-parametric operations such as an identity [24]. Differently, SNAS [22] uses the distribution of architectures as the main objective of the supernet's training DSNAS, and optimizes the likelihood of sampling good architectures. DSNAS [9] further discretizes SNAS, while adopting progressive early stopping. In this work, we seek to evaluate the impact of supernet splitting on these various one-shot NAS algorithms.

While one-shot NAS has made it possible to reduce search costs from several days down to a few hours, its performance is lackluster. It is exposed [23] that the architecture rankings produced by using a supernet proxy are highly degraded. This is largely attributed to co-adaptation [2]. Intuitively, the weights of each operation in the supernet after training converges are optimal in the context of joint training with every other operation in the space, but translate poorly to candidate architectures at evaluation time. This results in the mismatch between true and predicted architecture rankings.

### 2.2 Search space and benchmarks

The search space introduced in DARTS [15] remains the standard for dedicated one-shot NAS search spaces. In this search space, cells are the base unit of candidate architectures, and are represented as a directed acyclic graph, where the nodes represent intermediate features while the edges are selectable operations. Two types of cells are included, depending on whether the features are downsampled in the same layer. Replicas of the searched cells are then stacked on top of each other to form the full network (Figure 2). This structure puts high emphasis on searching the right operations, with little concern for the size of the architecture. The NAS-Bench-201 [6] space is a reduction of the DARTS space with a single type of searchable cell and a reduced number of selectable operations. Due to its simplicity, it has been possible to exhaustively evaluate all of its 15625 candidate architectures on three datasets : CIFAR-10, CIFAR-100 [13] and ImageNet16-120, which is a downscaled subset of ImageNet [5]. As such it has become an important benchmark for one-shot NAS methods, which we leverage to assess key aspects of the few-shot NAS framework.

### 2.3 Few-Shot NAS

Few-shot NAS [25] techniques are introduced to reduce co-adaptation in an orthogonal way to the one-shot search algorithm itself. Prior to the search, the supernet is splitted into several sub-supernets along an edge of the graph (Figure 1a), such that operations located on that edge are no longer trained jointly. This operation shrinks the search space and reduces the effects of co-adaptation. By applying it several times, a splitting tree is created, with sub-supernets as the resulting leaves. The final splits are affected by the branching factor, which corresponds to the number of groups created at each branch, and the depth of the tree, which is the number of

edges where splitting has been applied. The incurred time costs scale with the number of splits, unless an effective sub-supernet selection strategy is adopted. For this reason, the few-shot NAS paradigm can be seen as a middle-ground between the single-training procedure of one-shot NAS and the architecture sampling of traditional NAS. Using a naive policy where the supernet is split exhaustively along an edge, few-shot NAS reports an important gain to the correlation between supernet proxy and architecture ranking.

GM-NAS [8] expands on few-shot NAS with the proposal to group together operations whose standalone gradient is similar. They remark that different group of operations may suffer varying degrees of co-adaptation, depending on whether the gradients of the operations are pointing to similar directions. In order to measure whether this is the case, the gradient matching metric is introduced :

$$GM_k(o_i, o_j) = \mathcal{S}_{cos}[\nabla_w(\mathcal{L}(m_{o_i}^k, w)), \nabla_w(\mathcal{L}(m_{o_j}^k, w))] \tag{1}$$

where $\mathcal{S}_{cos}$ is the cosine similarity function, $\mathcal{L}$ is a loss function and $m_o^k$ is a sub-supernet with only operation $o$ left enabled on edge $k$.

After evaluating gradient matching for every pair of operations, groups can be formed via bruteforced min-cut optimization. Due to the lesser number of generated sub-supernets, deeper levels of the splitting tree can be considered. When combined with classic one-shot NAS algorithms, GM-NAS reports significant improvement, notably reaching near-optimality on all three dataset of NAS-Bench-201 and state-of-the-art performance in the DARTS space when used in conjunction with SNAS.

While GM-NAS introduces the question of searching for the best way to split a supernet, the underlying effects of splitting itself remain rather unexplored. Specifically, the results of the method could be attributed to the effectiveness of gradient matching as a metric, or to the positive influence of the splitting scheme. By generalizing the splitting framework further, we intend to give deeper insights into these techniques.

## 2.4 Zero-Shot NAS

In contrast with traditional NAS techniques and one-shot NAS, zero-shot NAS proposes to search for candidate architectures without training, relying instead on hand-crafted metrics to evaluate models at initialization within a single backward pass. This novel approach allows for low-cost, fast NAS that can be used in contexts such as edge computing [20]. Differently from their intended usage, we use metrics originated from this line of work to help with supernet splitting.

NASWOT [16] introduced this paradigm by greedily searching the space with a metric constructed from the covariance of the jacobian, which we hereafter refer to as `jacobcov`. Further zero-cost proxies were introduced in [1], which utilized the norm of the gradients `gradnorm` and several saliency metrics introduced in the neural network litterature : `snip`, `grasp` and `synflow`. These were used in conjunction with reinforcement learning or evolutionary algorithms.

Differently, TE-NAS [3] uses a mixture of metrics to prune a supernet down to a single path. The condition number of the Neural Tangent Kernel [10] and the number of linear regions are chosen to strike a balance of trainability and expressivity. In the following paragraphs, we use these metrics in a standalone way and refer to them as `ntk` and `#lr`.

The effectiveness of zero-shot NAS methods has been criticized, with subsequent work observing that naively counting the number of parameters in the model is a better proxy to model performance than most metrics employed in zero-shot NAS [12]. We include this metric in our study and refer to it as `#params`.

Further details about the zero-shot metrics can be found in Appendix B.

## 3 Methods

In summary, one-shot NAS methods train a supernet as a superposition of all candidate architectures in the space to guide architecture selection. Though, the joint training of all candidate architectures degrades the performance of the search. Few-shot NAS is introduced to mitigate this drawback by splitting the supernet into several sub-supernets, which are trained separately. The challenge of few-shot NAS is as follows : what is the best way to split the supernet? Motivated by the recent GM-NAS [8] which groups together architectures whose gradients are similar, we aim to extend the concept of metric-driven splitting to other promising metrics. In the following paragraphs, we introduce a further generalization of the GM-NAS framework to arbitrary network scoring metrics.

### 3.1 Supernet Splitting Framework

The effectiveness of GM-NAS [8] suggests that it is possible to learn how to split the original search space into subsets such that some of the sub-spaces will be more favorable to one-shot search. However, a single metric that does so has been introduced, with no guarantees to be the best metric for this task. Furthermore, the effects of splitting itself on the search are not well-enough known to ascertain that the choice of the metric is essential.

We seek to investigate the inner workings of supernet splitting. To do so, we introduce a supernet splitting framework inspired by GM-NAS. We generalize GM-NAS by using any metric that scores neural networks attributes to produce splits through pairwise comparison of each operation's impact on the metric. Intuitively, because these metrics are related to various aspects of a neural network (e.g. performance, trainability, expressivity), they can also be used to evaluate differences between several networks in terms of these aspects. Thus the relationship matrix of the edge can be constructed from :

$$\mathcal{M}_k(o_i, o_j) = \|f(m_{o_i}^k) - f(m_{o_j}^k)\|$$ (2)

where $f$ is a given metric and $m_o^k$ is a sub-supernet with only operation $o$ left enabled on edge $k$. Groups are formed from min-cut optimization over the one-dimensional metric axis operations lie on (Fig 1b).

In practice, metrics compute a score within a single forward and backward pass on a randomly initialized model with all but the target operation removed at the target edge. The similarity between each pair of operations with respect to the metric is the difference of the scores. Assuming a small enough number of operations in the space, this optimization is performed exhaustively by assessing the cost of every permutation, as in GM-NAS. Thus the optimization objective is similar :

$$\mathcal{U}_k = \arg\min_{\mathcal{U}_k \subseteq \mathcal{O}} \sum_{o \in \mathcal{U}_k, o' \in \mathcal{O} \setminus \mathcal{U}_k} \mathcal{M}_k(o, o')$$ (3)

where $\{\mathcal{U}_k, \mathcal{O} \setminus \mathcal{U}_k\}$ are the obtained operation partitions for edge $k$.

We implement the following metrics as the basis for supernet splitting: `gradientmatching`, `gradnorm`, `jacobcov`, `snip`, `grasp`, `synflow`, `ntk`, `#lr`, `#params`.

### 3.2 Splitting Setup

We generate sub-supernets for every metric in the NAS-Bench-201 space, based on the same supernet checkpoint. Training details for the supernet can be found in Appendix A.2. The splitting tree goes to a depth of 3, with operations being split into two groups at each branch. In the end, we obtain 8 sub-supernet for each metric.

Remark that the generation of a splitting tree supports other hyperparameters than the metric alone. The number of groups at each branch, also referred to as branching factor [8], is an extra hyperparameter, which can degenerate the method to vanilla few-shot NAS [25] when set to its

| CIFAR10 | | | | |
|---|---|---|---|---|
| Metric | One-shot NAS algorithms | | | |
| | DARTS-1st | DARTS-2nd | SNAS | DSNAS |
| no splitting | 70.92 | 91.52 | 93.66 ±0.08 | 92.10 ±0.13 |
| random | 88.71 | 92.96 | 93.74 ±0.03 | 93.88 ±0.34 |
| gradientmatching | 92.04 | 93.31 | **94.13** ±0.30 | 94.22 ±0.00 |
| gradnorm | 91.89 | **93.66** | 93.61 ±0.11 | 93.34 ±0.19 |
| jacobcov | 91.86 | 93.34 | 93.61 ±0.11 | 93.91 ±0.05 |
| snip | **92.17** | 93.36 | 93.92 ±0.32 | 93.28 ±0.13 |
| grasp | **92.17** | 92.69 | 93.57 ±0.09 | 93.46 ±0.21 |
| synflow | 86.51 | 92.84 | 93.76 ±0.00 | 93.86 ±0.27 |
| ntk | 90.44 | 93.59 | 93.71 ±0.00 | 93.53 ±0.26 |
| #lr | 86.51 | 92.94 | 93.76 ±0.00 | **94.36** ±0.00 |
| #params | 91.88 | 92.71 | 93.91 ±0.32 | 94.30 ±0.09 |

Table 1: Test accuracy of the best found architecture across 8 sub-supernets on CIFAR10

| CIFAR100 | | | | |
|---|---|---|---|---|
| Metric | One-shot NAS algorithms | | | |
| | DARTS-1st | DARTS-2nd | SNAS | DSNAS |
| no splitting | 38.97 | 55.70 | 70.91 ±0.00 | 58.20 ±0.25 |
| random | 61.29 | 68.42 | 71.56 ±1.07 | 71.75 ±1.25 |
| gradientmatching | 68.55 | **72.75** | 71.59 ±1.05 | **73.51** ±0.00 |
| gradnorm | 67.63 | 70.59 | 70.87 ±0.40 | 68.47 ±0.00 |
| jacobcov | 68.10 | 69.43 | 70.84 ±0.43 | 71.15 ±0.00 |
| snip | **69.24** | 69.52 | 71.15 ±0.00 | 68.07 ±0.28 |
| grasp | 67.89 | 69.95 | 70.84 ±0.43 | 68.47 ±0.00 |
| synflow | 67.16 | 70.35 | 71.60 ±0.35 | 73.26 ±0.35 |
| ntk | 67.89 | 71.42 | 70.54 ±0.43 | 71.15 ±0.00 |
| #lr | 67.75 | 69.39 | **71.85** ±0.00 | **73.51** ±0.00 |
| #params | 58.31 | 69.60 | 70.37 ±0.49 | 72.23 ±0.86 |

Table 2: Test accuracy of the best found architecture across 8 sub-supernets on CIFAR100

upper limit. This parameter could also be controlled more finely at each level of the tree with a dedicated policy. For the sake of simplicity, we fix the branching factor in subsequent experiments and only split operations into two groups. However, we do not apply the strict condition to keep splits balanced as in GM-NAS. We argue that some operations may behave so differently from others with respect to a specific metric on a given edge that it is justified to separate them from the others, even if this results in an uneven split.

Differently from GM-NAS, we do not select an edge to split at run time, instead opting to always split the first three edges of the supernet in that order. In a framework with edge selection, various metrics may result in different edges being selected, thus keeping splitted edges fixed ensures full comparability between the metrics and keeps the focus on their operation selection capabilities. Furthermore, we do not apply warmup to sub-supernets in between splits, as the weights and gradients of the supernet, which all metrics are based on, have already converged. These weights are instead restarted once all splitting has concluded to avoid bias to the performance of associated one-shot NAS algorithms.

## 4 Analysis of supernet splitting

### 4.1 Impact of splitting

While the effectiveness of few-shot NAS approaches over regular one-shot NAS has been shown in previous work, it remains unclear what the impact on performance of splitting is compared to the

| | ImageNet16-120 | | | |
|---|---|---|---|---|
| Metric | One-shot NAS algorithms | | | |
| | DARTS-1st | DARTS-2nd | SNAS | DSNAS |
| no splitting | 18.41 | 38.39 | 46.34 ±0.00 | 28.26 ±0.35 |
| random | 39.08 | 43.23 | 46.74 ±0.40 | 45.35 ±0.84 |
| gradientmatching | 40.02 | **45.48** | **47.31** ±0.00 | 46.07 ±0.19 |
| gradnorm | 39.90 | 44.60 | 44.23 ±0.00 | 43.25 ±1.71 |
| jacobcov | **41.00** | 43.03 | 44.60 ±0.26 | 44.23 ±0.00 |
| snip | 36.60 | 43.77 | 44.23 ±0.00 | 43.99 ±1.76 |
| grasp | 39.90 | 44.17 | 44.23 ±0.00 | 45.23 ±0.00 |
| synflow | 27.88 | 39.33 | 46.85 ±0.00 | 46.68 ±0.24 |
| ntk | **41.00** | 42.90 | 44.41 ±0.26 | 44.23 ±0.00 |
| #lr | 27.88 | 42.77 | 46.85 ±0.00 | **46.85** ±0.00 |
| #params | 39.08 | 43.72 | 46.85 ±0.00 | 46.34 ±0.00 |

Table 3: Test accuracy of the best found architecture across 8 sub-supernets on ImageNet16-120

choice of the metric. In this section, we evaluate the splits generated with 9 different metrics on 4 different one-shot NAS algorithms. In order to assess the impact of splitting by itself against the metrics, we also compare them against the random baseline, where splits are generated by sorting operations randomly into 2 groups. For a fairer comparison, we report SNAS and DSNAS runs with the same 3 seeds. In the case of DARTS algorithms, previous work [6] has shown that these algorithms are fully deterministic in the NAS-Bench-201 search space, therefore conducting repeat experiments is less relevant. We conduct these experiments on all 3 datasets of NAS-Bench-201 and report the results in Table 1, Table 2, and Table 3. Note that we report only the performance of the best architecture found collectively by all sub-supernets, which is indicative of the maximum potential of each split-algorithm pair. The search cost required to reach this potential is the cost of running the one-shot NAS on all sub-supernets. Reducing this cost via e.g. supernet selection strategies may yield lower performance. Our experimental setup for each algorithm can be found in appendix A.

Our results confirm the positive impact of splitting on the performance of various one-shot NAS algorithms. We observe that for most one-shot NAS algorithms, a better architecture is found when splitting regardless of the metric used. Note that even for the SNAS [22] algorithms that converge to a good architecture without splitting, performance is seldom degraded after splitting. Therefore, obtained results reinforce the core assumption of the few-shot NAS framework [8, 25], that partitioning the search space is beneficial to the performance of conjoined NAS algorithms.

As for the impact of the metrics used to generate sub-supernet networks, results reported in the different tables show that partitioning operations randomly yields similar or better results compared with other metrics. Furthermore, we find that the optimal metric varies across different datasets and different algorithms. In many cases, this happens with low variance, indicating that the splits generated by the metric are especially well suited for this combination of dataset and algorithm. Intuitively, the accuracy of architectures in the space varies relative to each other as tasks become more complex, while inherent properties of the supernet picked up by the selected metrics may not change as much. This indicates that the choice of an appropriate metric is not as important as the splitting operation itself. However, the fact that an algorithm searches more effectively when the space is split in specific ways can be linked with how the architectures are distributed within the splits. Therefore, we take a deeper look at sub-space distributions in the following section.

## 4.2 What is in a split?

The contents of each split remain unexplored in previous works. Indeed, both few-shot NAS and GM-NAS select a single sub-supernet based on its validation accuracy to cut costs down to similar

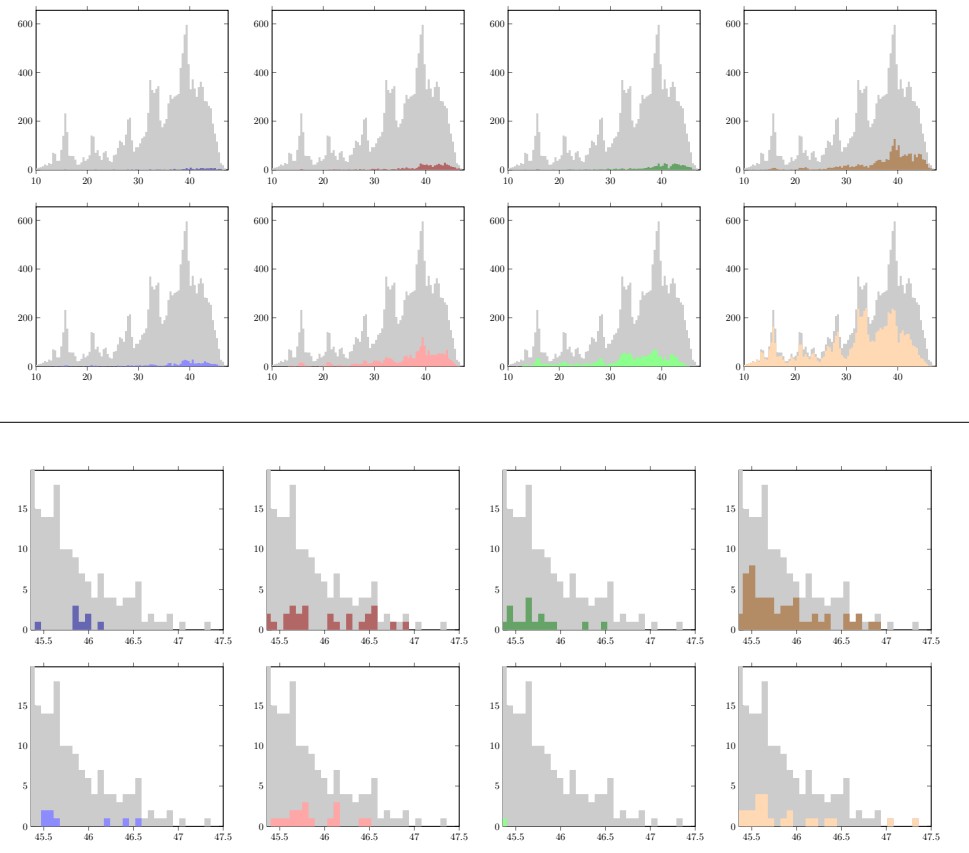

Figure 3: Distribution of NAS-Bench-201 [6] architecture performance on ImageNet16-120 across sub-spaces generated with `gradientmatching`. Each distribution represents all architectures contained in a single sub-space. Overall distribution appears in gray.
(Top) : full distribution. (Bottom) : distribution restricted to top 1% architectures in the space.

levels to regular one-shot NAS. However, this disregards the fact that good architectures may be located in more than one split. Moreover, the distribution of architectures in the splits could explain why some metrics are more advantageous for particular algorithms. Specifically, the ideal split would group architectures based on their performance, with all best-performing architectures located in a single split.

For each metric and dataset, we use the same 8 splits described in 3.2 and plot the distribution of architectures in each split with respect to their test accuracy from the tabulated NAS-Bench-201 [6] benchmark. Due to the sub-spaces being disjointed, the union of all 8 distributions amounts to the full search space. An example is shown in Fig 3 (top). However, the vast majority of architectures in the search space have mediocre performance, while NAS algorithms will mostly converge towards the best architectures. Therefore, looking at the full distribution gives no insight on whether a good architecture is likely to be found in that sub-space. Thus, we display a second view of the distribution, restricted to the top 1% of architectures in the search space. An example is represented in Fig 3 (bottom). We refer to Appendix D for distribution plots with other metrics and datasets.

We observe that metrics create splits following two types of behaviors. Splitting with `gradnorm`, `jacobcov`, `snip`, `grasp` or `ntk` creates balanced sub-spaces following closely the distribution of the original space. Meanwhile, splitting with `gradientmatching`, `synflow`, `#lr` or `#params` groups a small number of architectures in some spaces, and a number roughly equal to half the size of the full space in another space. Importantly, we observe that across all metrics, the top 1% of architectures

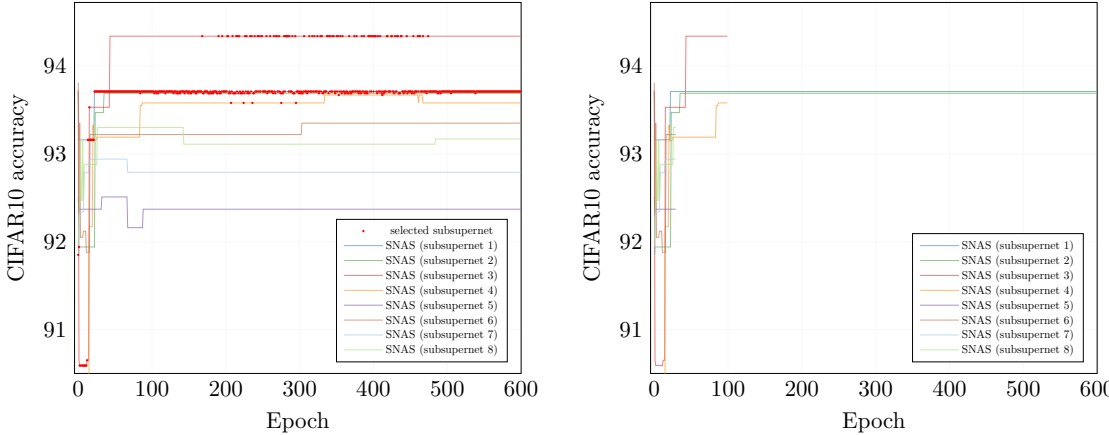

Figure 4: Search phase of SNAS [22] with 8 sub-supernets, splits obtained with `gradientmatching`.
(Left) Test accuracy of discovered candidate architecture at every epoch of the training. Red dots denote the sub-supernet with best validation accuracy for the current epoch.
(Right) Successive halving simulation. At epochs 30 and 100, the bottom half of sub-supernets with the lowest validation accuracy is discarded.

in the space is located in several sub-spaces, which is suboptimal. Unsurprisingly, when a larger proportion of the space's architecture is located in a single sub-space, a larger number of top 1% architectures is also present. Therefore, while some metrics may separate architectures in more advantageous ways, the performance gain could simply be attributed to the search taking place in a smaller search space, rather than inherent co-adaptation reduction inside the sub-supernets. Moreover, top architectures being spread out across multiple sub-spaces makes supernet selection impractical, as one may simply discard the global optimum while doing so.

## 4.3 Evaluating supernet selection

While the performance ceiling of NAS with splitting can be reached by training all sub-supernets exhaustively, past works strive to reduce the search down to similar costs as regular one-shot NAS. Few-shot NAS [25] selects the top $k$ sub-supernets with the highest validation accuracy after copying weights over from the source supernet. GM-NAS [8] propose a successive halving approach inspired by bandit optimization [11, 14] and discard the bottom half of architectures based on their validation accuracy following a set schedule, with a 2.5x cost increase only for 8 sub-supernets, compared to one-shot NAS. Interestingly, both approaches use sub-supernet accuracy as a proxy to being a good supernet, but few guarantees of this relationship have been given. In order to assess whether accuracy is a good proxy for supernet selection, we use splits obtained with `gradientmatching` and run SNAS [22] on each of 8 sub-supernets, using the same experimental setup described in (appendix), only extended to 600 epochs.

We report the results for CIFAR-10 in Fig 4. Firstly, we find that, by the end of 600 epochs and for most of the training, the sub-supernet with highest validation accuracy is not the one that finds the highest performing architecture with SNAS, which shows that validation accuracy is not a suitable proxy to selecting a sub-supernet. Furthermore, by applying a successive halving scheme similar to GM-NAS, where half of the sub-spaces are trimmed at epochs (30, 100, 600), we observe that the best sub-space is discarded early on. This indicates that supernet accuracy is especially inaccurate as a proxy in early stages of the training. As such, supernet selection with validation accuracy is highly likely to degrade the final performance of few-shot NAS compared to its maximum possible performance.

## 5 Conclusion

One-shot NAS methods have been proposed to reduce the cost of running NAS, but they often degrade the performance of the search. Few-shot NAS methods tackle this issue by splitting the supernet into sub-supernets trained separately, each with a reduced level of weight-sharing. The challenge is to find the best way to split the supernet. A significant contribution in this direction is GM-NAS, which utilizes a gradient matching score to group operations in a splitting schema rather than an exhaustive approach. We extend this splitting framework to support any given splitting metric. By combining this generalization with several architecture scoring metrics from the zero-shot NAS literature, we can generate a greater variety of sub-spaces. We observe that partitioning the search space invariably increases the performance of one-shot NAS algorithms, while the best metric may change unpredictably depending on the task and the conjoined algorithm. Therefore, splitting itself is more important than the choice of a splitting metric. Moreover, we show that top architectures in the space are spread out across multiple sub-spaces, making it difficult to determine which sub-space is the best. Finally, we evaluate supernet selection techniques and determine that sub-supernet accuracy is an inadequate proxy for finding a good sub-space and might only degrade the performances. Overall, our work shows that, while few-shot NAS is a promising line of work to boost NAS performances in the one-shot setting, it requires careful tuning to reach its maximum performance with reduced costs.

## 6 Broader Impact Statement

After careful reflection, the authors have determined that this work presents no notable negative impacts to society or the environment.

**Acknowledgements**. This work was granted access to the HPC resources of IDRIS under the allocation AD011013972 made by GENCI.

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

## A  Hyperparameters

In this section, we detail the hyperparameters used in our experiments.

### A.1  Model hyperparameters

The models used in our work follow the NAS-Bench-201 space described in section 2.2, with the following parameters: 16 initial channels, 2 reductions with 5 cells in-between each reduction, for a total depth of 17 cells.

### A.2  Supernet training hyperparameters

The trained supernet is used as the basis for splitting. It is trained by optimizing the cross-entropy loss via SGD with batch size 128, initial learning rate 0.1, momentum 0.9 and weight decay 0.0005. The learning rate is annealed down to 0 following a cosine scheduler over the course of 200 epochs.

### A.3  DARTS hyperparameters

We follow the same hyperparameters to conduct the search over DARTS [15] 1st and 2nd order. Half of the training examples are held out as the validation dataset for the search. We train the supernet parameters by optimizing the cross-entropy loss using Nesterov SGD for 50 epochs with batch size 64, initial learning rate 0.025, momentum 0.9 and weight decay 0.0003. The learning rate is annealed down to 0.001 using a cosine scheduler. For the training of the architecture parameters, we optimize them with Adam optimizer, fixed learning rate 0.0003 and weight decay 0.001.

### A.4  SNAS hyperparameters

We detail the hyperparameters for the search over SNAS [22]. Half of the training examples are held out as the validation dataset for the search. In order to speed up the experiments, we use the same hyperparameters as in DARTS A.3 for both the supernet parameters and architecture parameters. As for the temperature, it is annealed from 1.0 to 0.1 following the cosine schedule. We conduct the training over 50 epochs. Note that this is significantly less than the 600 epochs prescribed for SNAS [22]. Empirically, we find that by 50 epochs, most sub-supernets reach their peak performance or convergence point (cf. Figure 4).

### A.5  DSNAS hyperparameters

We detail the hyperparameters for the search over DSNAS [9]. Once more, half of the training examples are held out as the validation dataset for the search. AWe use the same hyperparameters as DARTS and DSNAS A.4 for supernet parameters and architecture parameters. For the progressive early stopping, we follow the same condition as described in DSNAS and set a threshold of 0.8. We conduct the search for 50 epochs.

## B  Details of the metrics

In this section, we document the definition of each metric used.

### B.1  `gradnorm`

Gradients of the model are computed over a single batch of training data, then concatenated. This metric is equal to the norm of the obtained tensor :

$$\texttt{gradnorm}(m) = \|\nabla_w(\mathcal{L}(m, w))\|_{\mathcal{F}} \tag{4}$$

where $m$ is the evaluated model with parameters $w$, $\mathcal{L}$ is the cross-entropy loss and $\|\cdot\|_{\mathcal{F}}$ is the Frobenius norm.

### B.2  `jacobcov`

As with `gradnorm`, gradients are computed over a single batch of training data then concatenated. We first define the covariance matrix of obtained Jacobian tensor:

$$C(m) = E[(\nabla_w(\mathcal{L}(m, w) - E(\nabla_w(\mathcal{L}(m, w)))^T(\nabla_w(\mathcal{L}(m, w) - E(\nabla_w(\mathcal{L}(m, w)))] \tag{5}$$

where $m$ is the evaluated model with parameters $w$ and $\mathcal{L}$ is the cross-entropy loss.

Let $\lambda_1^m \dots \lambda_n^m$ be the eigenvalues of $C(m)$. The `jacobcov` metric is defined as follows:

$$\texttt{jacobcov}(m) = -\sum_{i=1}^{n}(log(\lambda_i^m) + \frac{1}{\lambda_i^m}) \tag{6}$$

### B.3  `snip`

The `snip` metric is computed over single batches of training data and is defined as follows:

$$\texttt{snip}(m) = \sum_i^N |\frac{\partial \mathcal{L}}{\partial \theta_i} \odot \theta_i| \tag{7}$$

where $i$ are the layers of the model, with $N$ the total number of layers. $\theta_i$ are the parameters at layer $i$ and $\mathcal{L}$ is the cross-entropy loss.

### B.4  `grasp`

The `grasp` metric is computed over single batches of training data and is defined as follows:

$$\texttt{grasp}(m) = \sum_i^N -(H\frac{\partial \mathcal{L}}{\partial \theta_i}) \odot \theta_i \tag{8}$$

where $i$ are the layers of the model, with $N$ the total number of layers. $\theta_i$ are the parameters at layer $i$ and $\mathcal{L}$ is the cross-entropy loss. $H$ is the Hessian, which is estimated following [18].

### B.5  `synflow`

The `synflow` metric is similar to `snip`:

$$\texttt{synflow}(m) = \sum_i^N |\frac{\partial \mathcal{L}}{\partial \theta_i} \odot \theta_i| \tag{9}$$

However, it is not computed over batches of training data. Instead, a batch of synthetic data with full matrices of 1s is passed to the model.

### B.6 `ntk`

The `ntk` metric is the condition number of the Neural Tangent Kernel (NTK) [10].

Gradients are computed over a single batch of training data then concatenated. Then, following TE-NAS [3], the NTK is approximated as:

$$\Theta(m) = \nabla_w(\mathcal{L}(m, w))^T \nabla_w(\mathcal{L}(m, w)) \tag{10}$$

where $m$ is the evaluated model with parameters $w$ and $\mathcal{L}$ is the cross-entropy loss. Let $\lambda_0^{\Theta(m)} \ldots \lambda_n^{\Theta(m)}$ the eigenvalues of $\Theta(m)$ sorted in ascending order of magnitude, the condition number of the NTK is defined as:

$$\text{ntk}(m) = \lambda_n^{\Theta(m)} / \lambda_0^{\Theta(m)} \tag{11}$$

### B.7 `#lr`

The `#lr` metric is defined as the number of linear regions after the activations of the model. In practice, the number of linear regions is estimated from a batch of training data, through careful analysis of the signs of the post-activations, following TE-NAS [3].

### B.8 `#params`

The `#params` metric is simply defined as the number of parameters in the model, which is obtained at initialization with no training data needed.

## C Ablation studies on architecture selection

In this section, we report the results of the experiment from section 4.3 when splitting using various other metrics. All other parameters of the experiment remain the same, and we use SNAS [22] on the CIFAR10 task for the search.

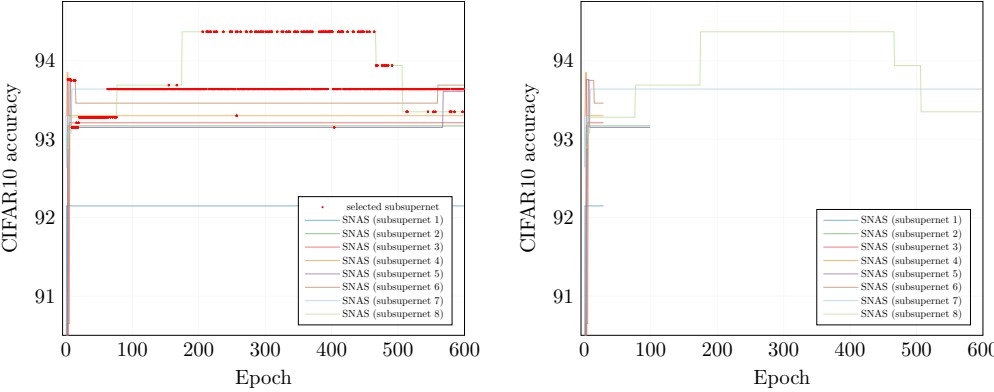

Figure 5: Search phase of SNAS with 8 sub-supernets, splits obtained with `gradnorm`.

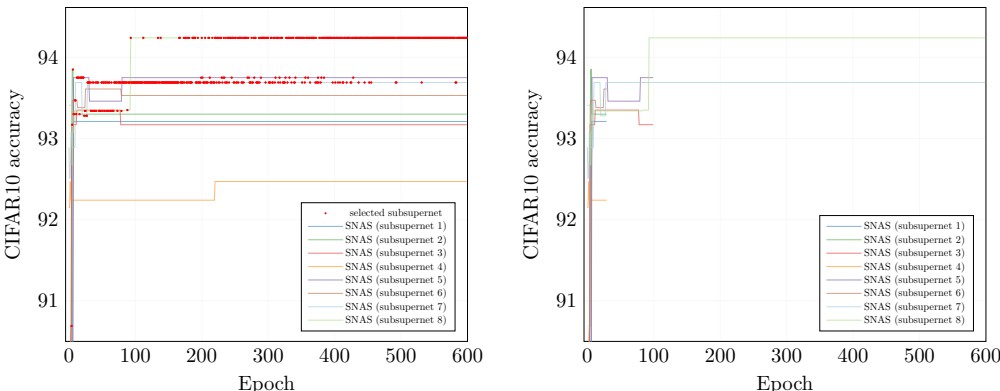

Figure 6: Search phase of SNAS with 8 sub-supernets, splits obtained with `jacobcov`.

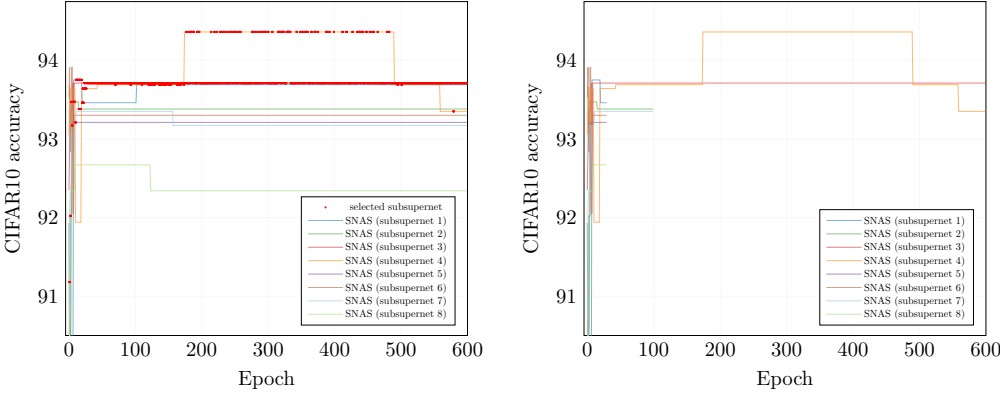

Figure 7: Search phase of SNAS with 8 sub-supernets, splits obtained with `ntk`.

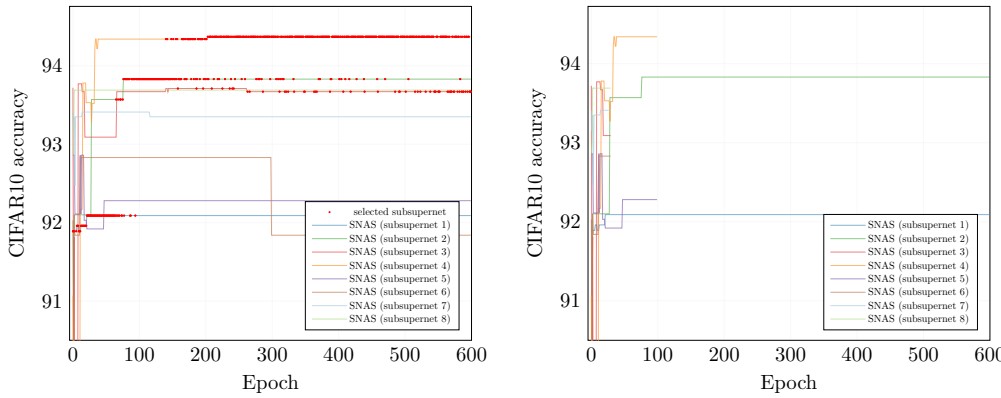

Figure 8: Search phase of SNAS with 8 sub-supernets, splits obtained with `params`.

Out of 5 reported metrics including `gradientmatching`, `gradnorm`, `jacobcov`, `ntk` and `params`, only the supernet selection phase of `jacobcov` yields satisfactory results, where the sub-supernet that finds the best performing architecture is also clearly the one selected in the end, while not being cut prematurely if applying successive halving.

We observe in the case of `params` that supernet selection remains very unstable up to the end of the training, with no sub-supernet coming out as the clear winner, while the best performing sub-supernet would be cut by applying successive halving.

Finally, the cases of `gradnorm` and `ntk` show that best-performing sub-supernets can experience collapse towards the end of the training, resulting in the selection of a sub-optimal supernet compared to the global peak over the search. This could indicate a mismatch of the search algorithm with supernet selection.

Overall, supernet selection using validation accuracy as a proxy often exhibits undesirable behaviors for finding the best performing sub-supernet.

# D Architecture distributions

## D.1 CIFAR-10

### D.1.1 `gradientmaching`.

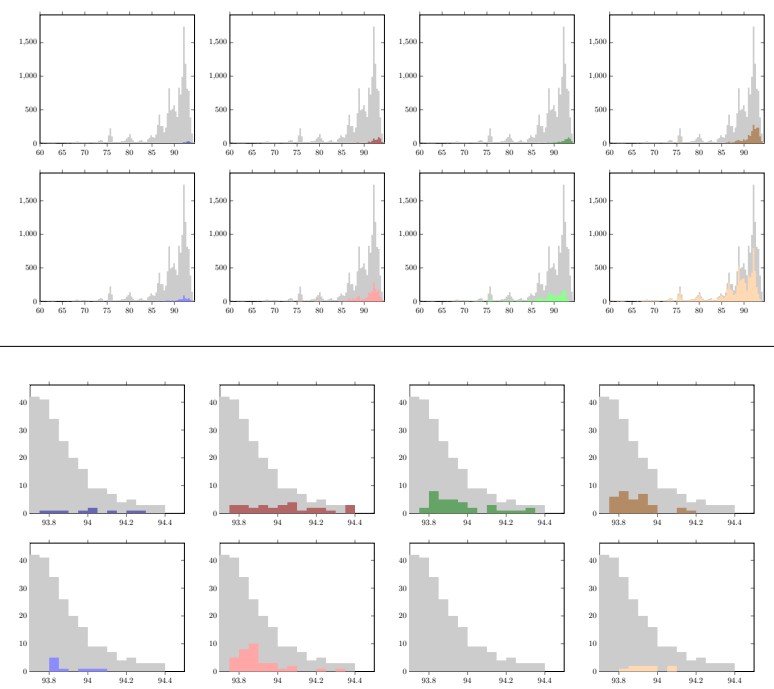

### D.1.2 `gradnorm`.

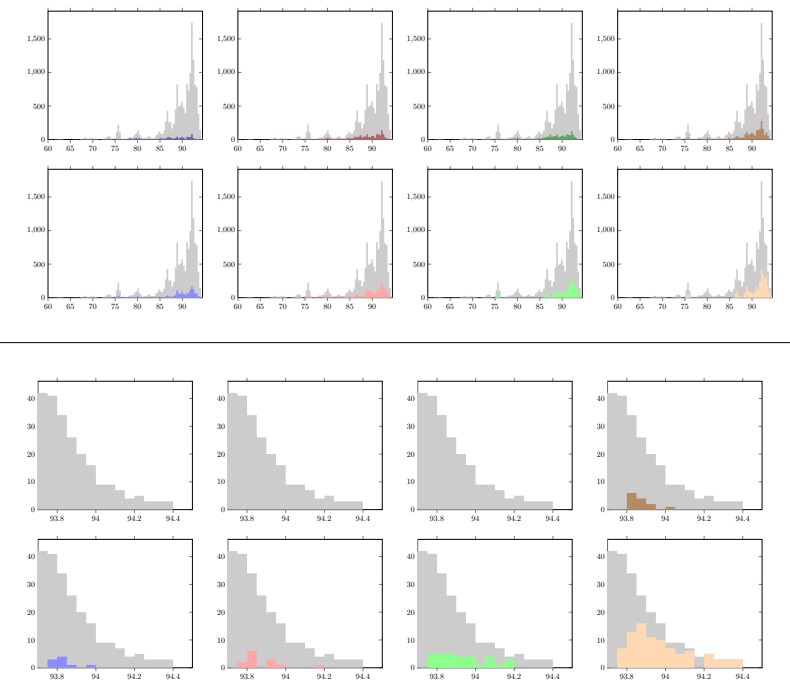

### D.1.3 `jacobcov`.

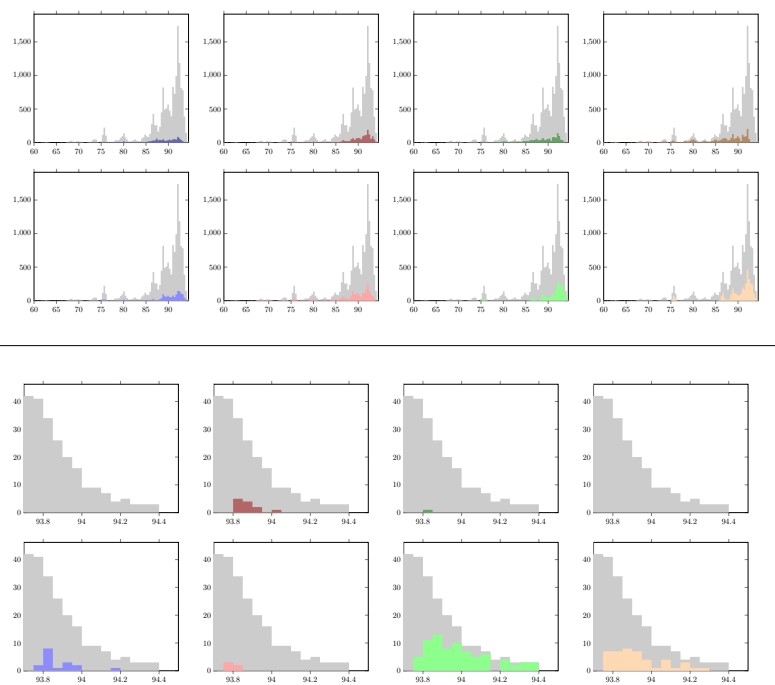

### D.1.4 `snip`.

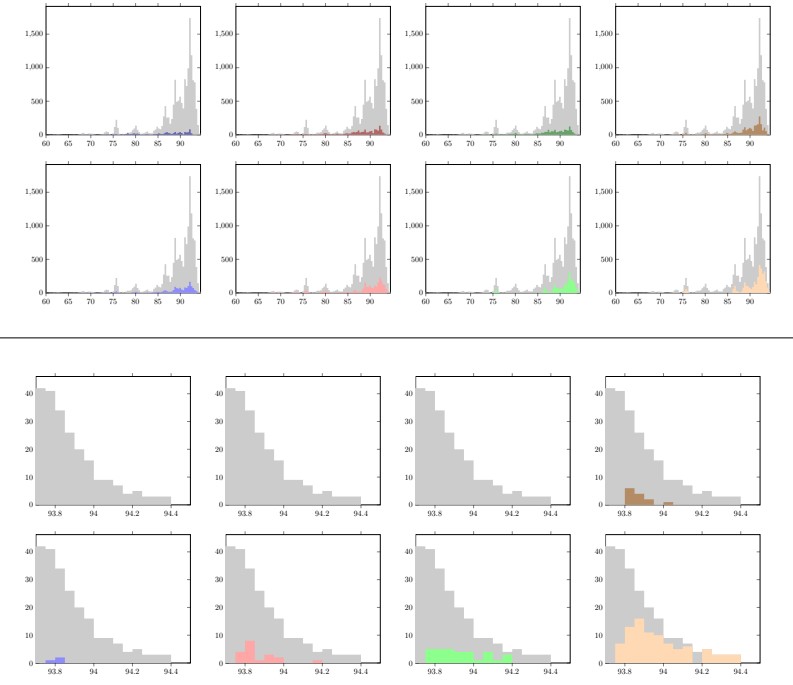

### D.1.5 `grasp`.

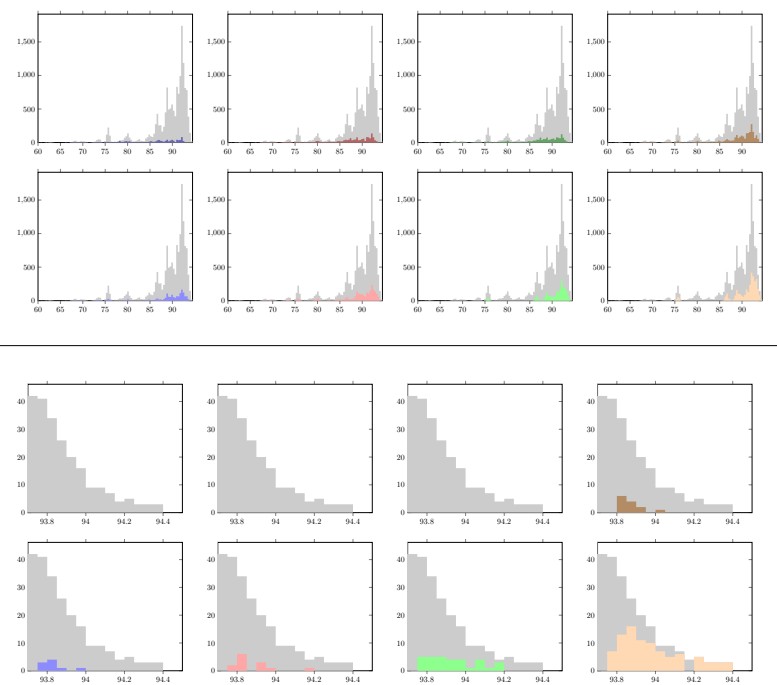

### D.1.6 `synflow`.

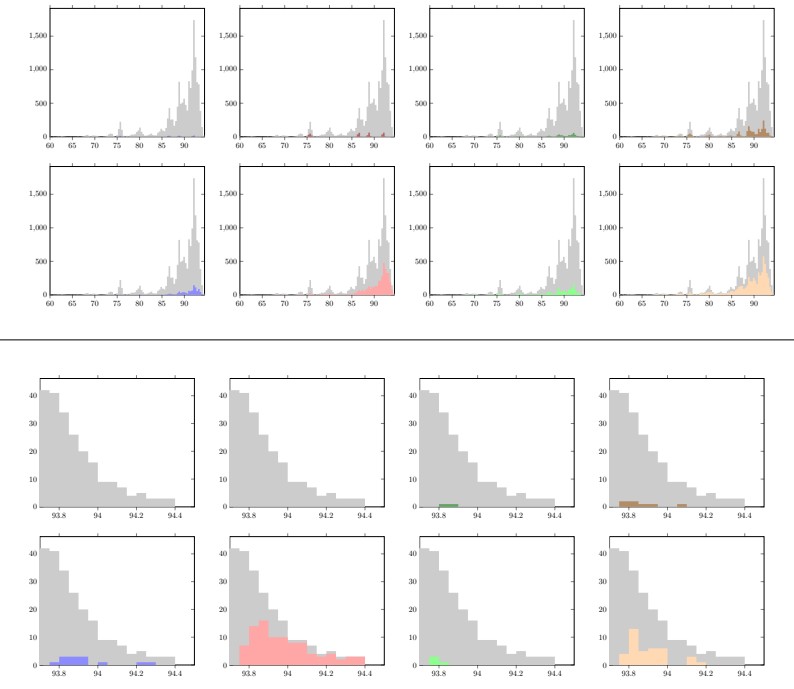

### D.1.7 `ntk`.

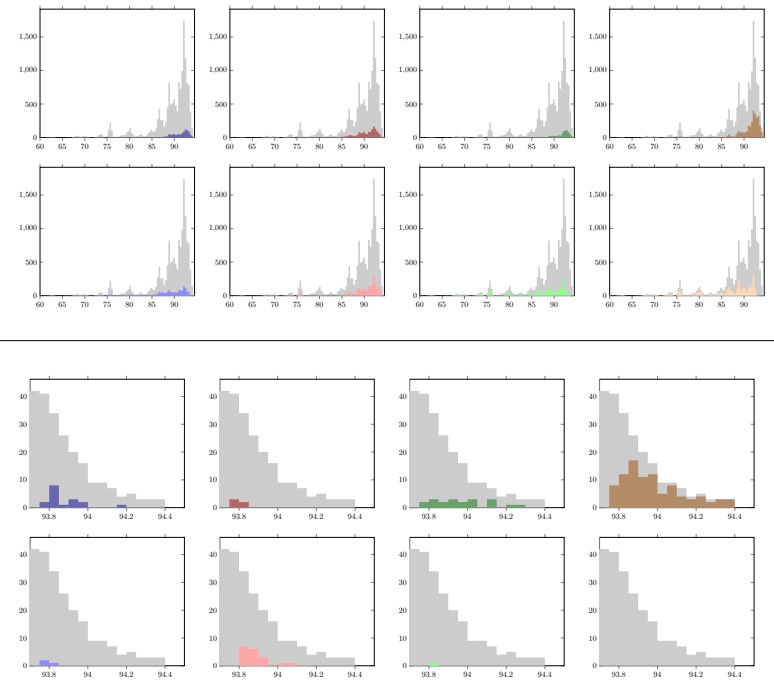

### D.1.8 `#lr`.

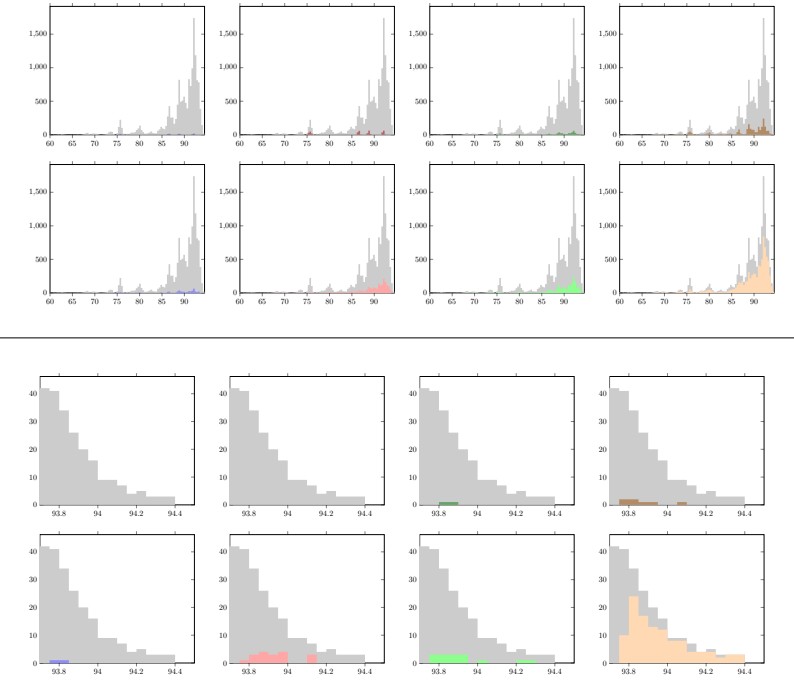

### D.1.9 `#params`.

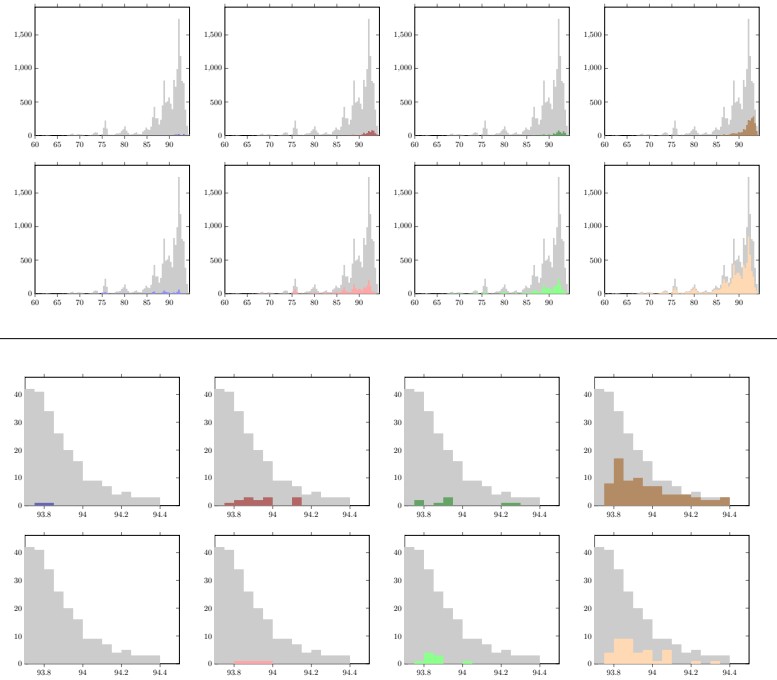

### D.2 CIFAR100

### D.2.1 `gradientmatching`.

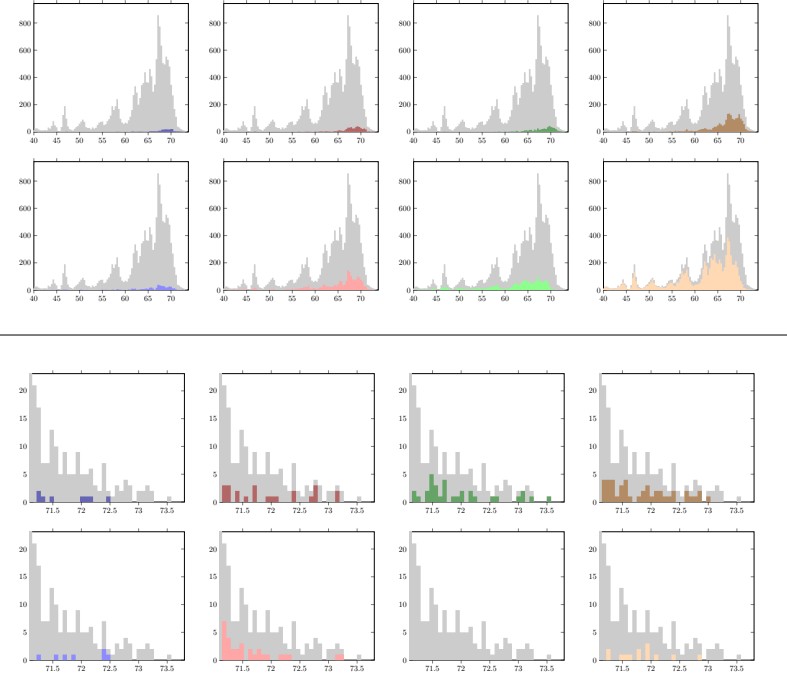

### D.2.2 `gradnorm`.

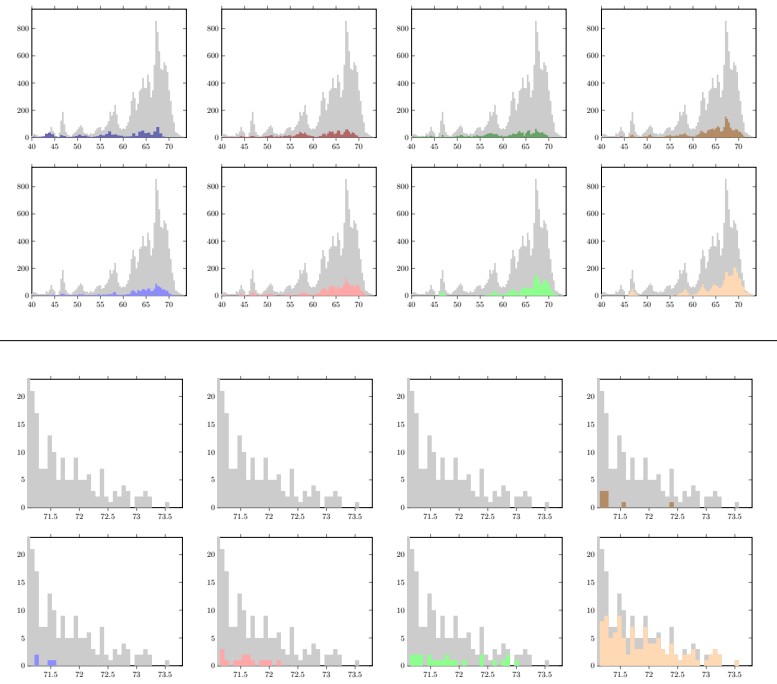

### D.2.3 `jacobcov`.

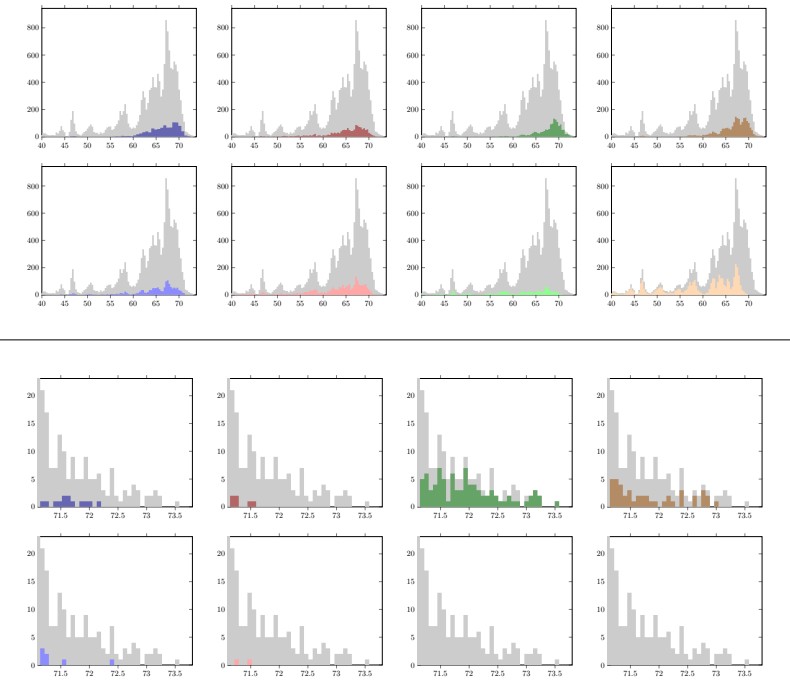

### D.2.4 `snip`.

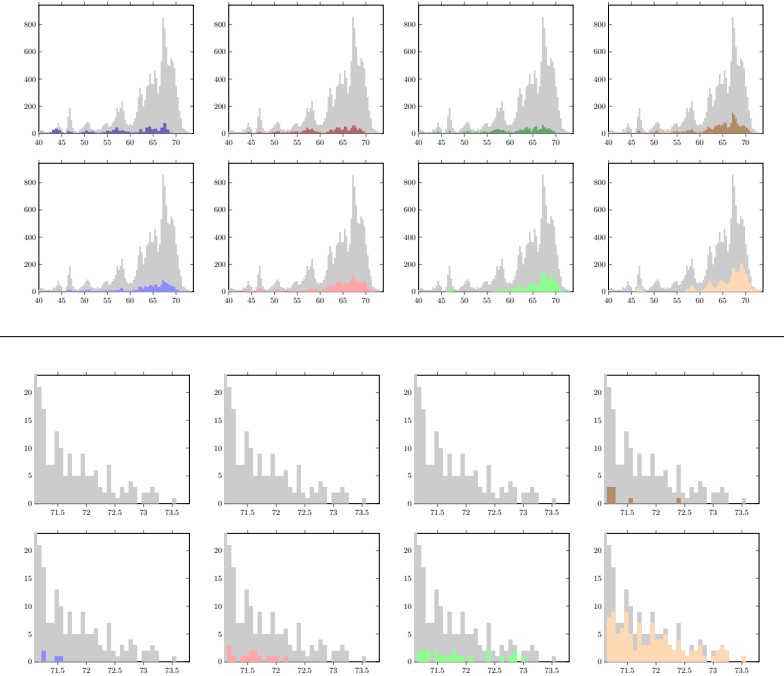

### D.2.5 `grasp`.

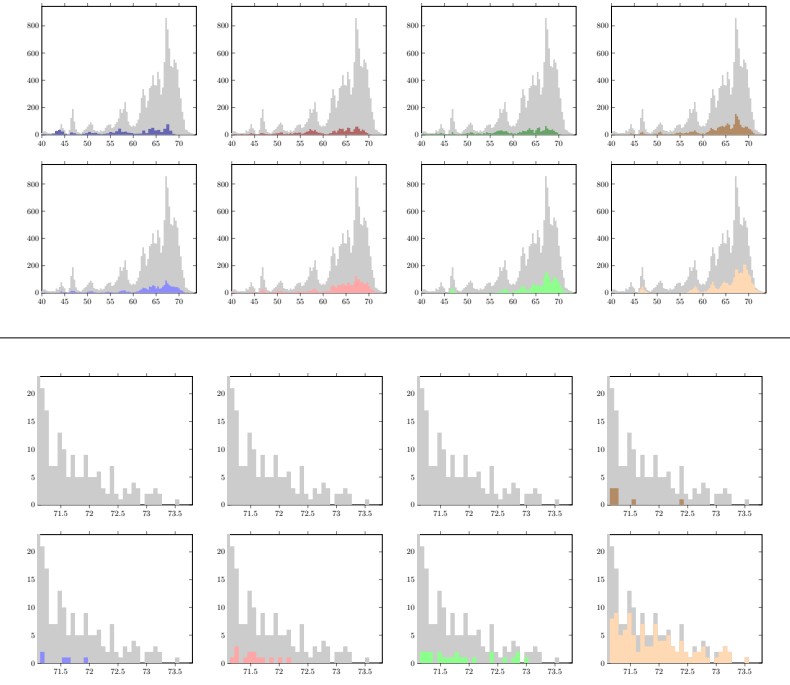

### D.2.6 `synflow`.

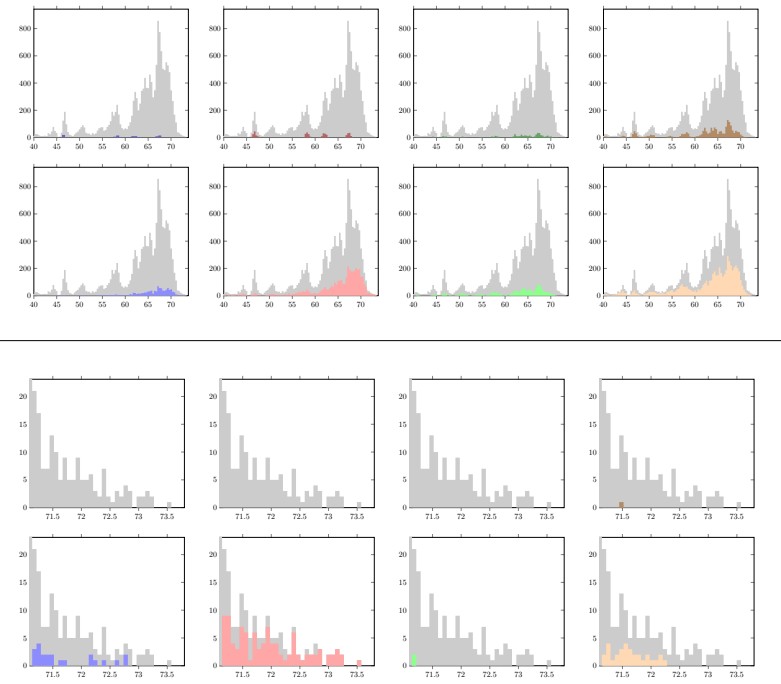

### D.2.7 `ntk`.

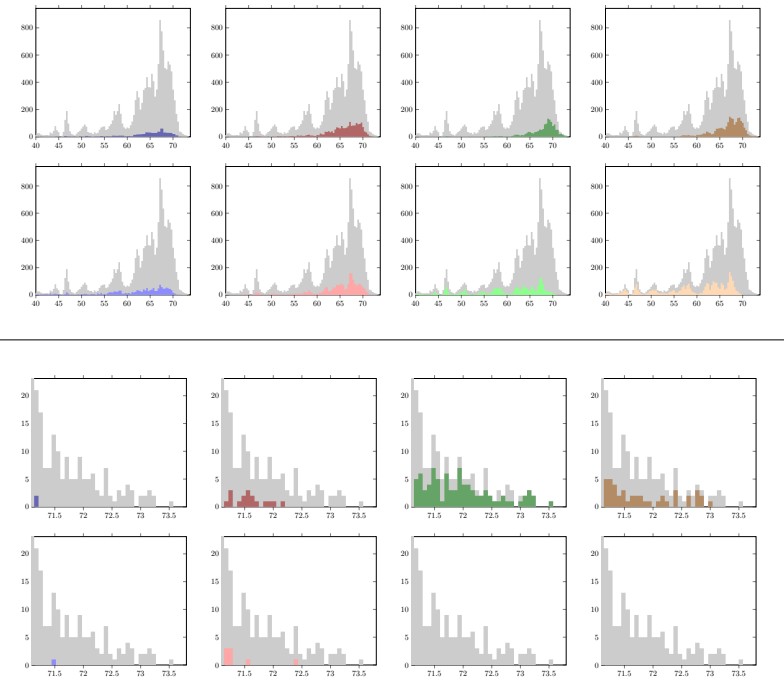

## D.2.8 `#lr`.

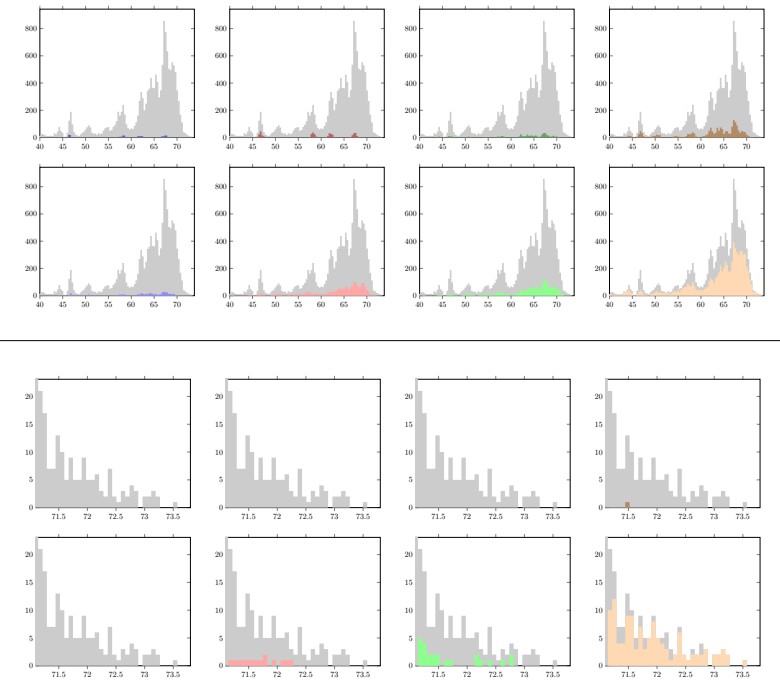

## D.2.9 `#params`.

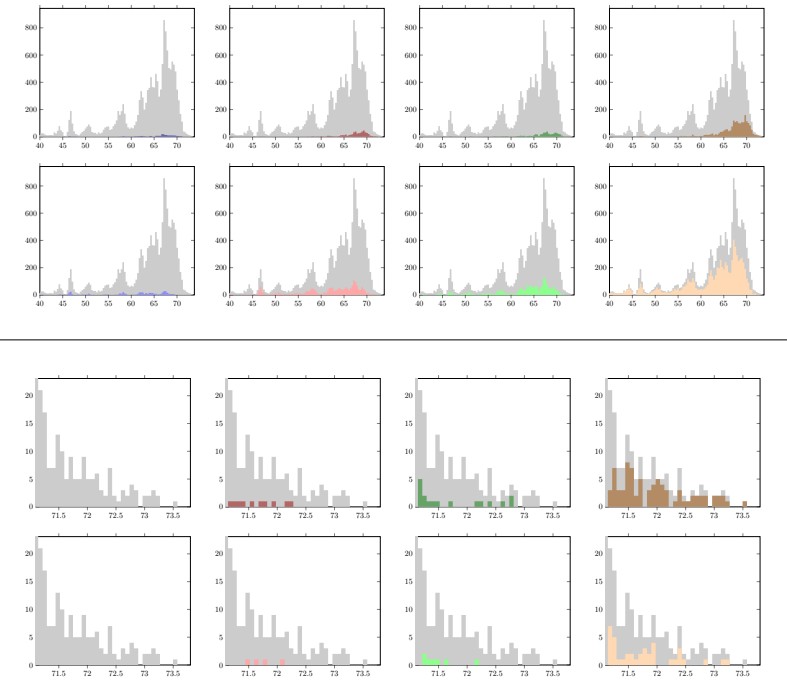

## D.3 ImageNet16-120

### D.3.1 `gradientmatching`.

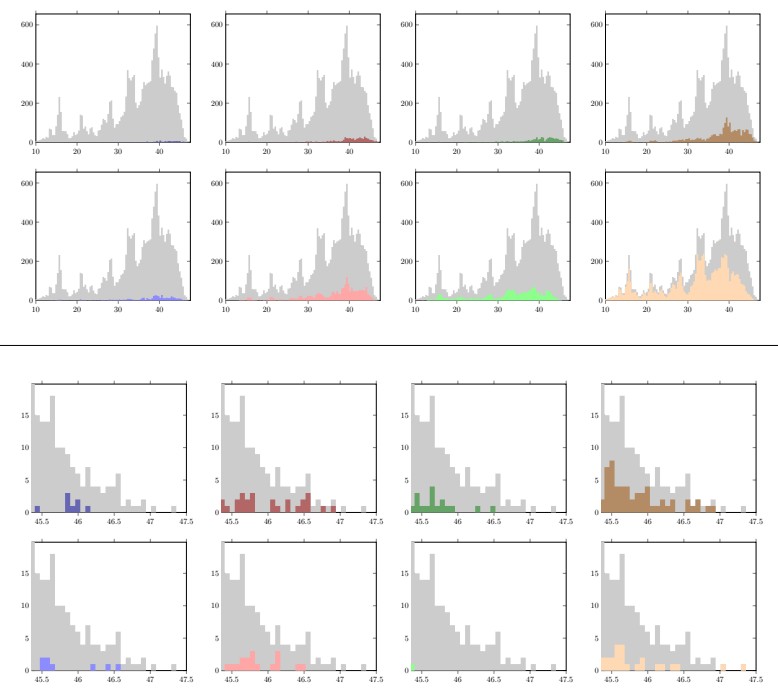

### D.3.2 `gradnorm`.

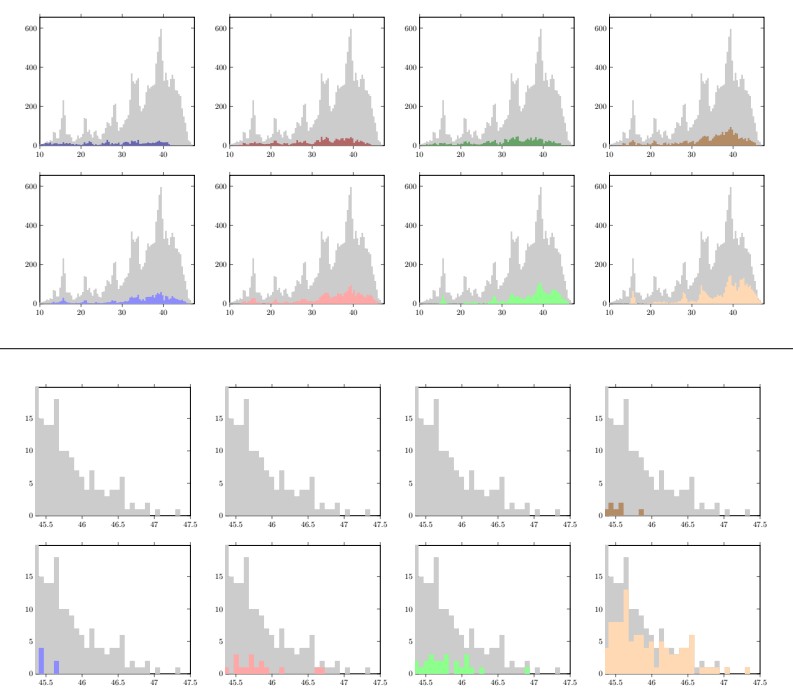

### D.3.3 `jacobcov`.

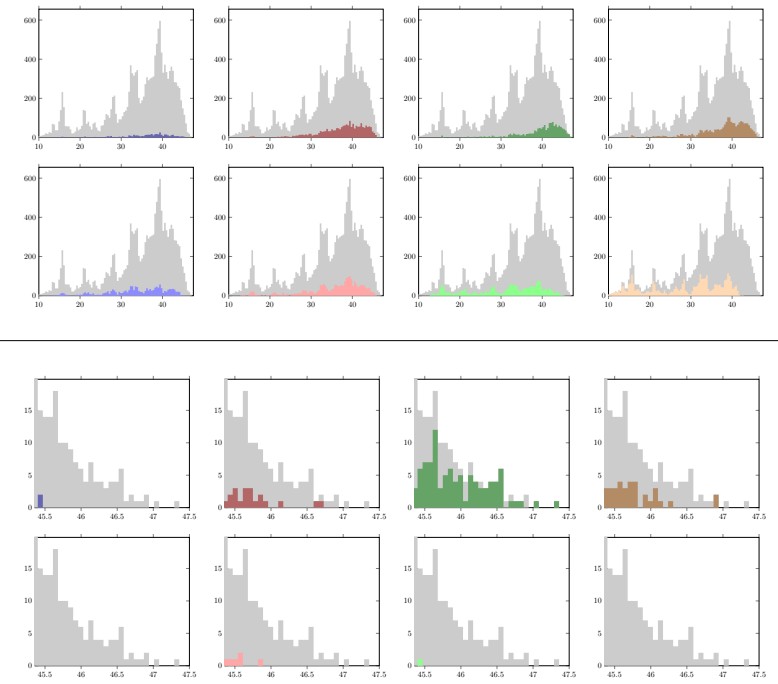

### D.3.4 `snip`.

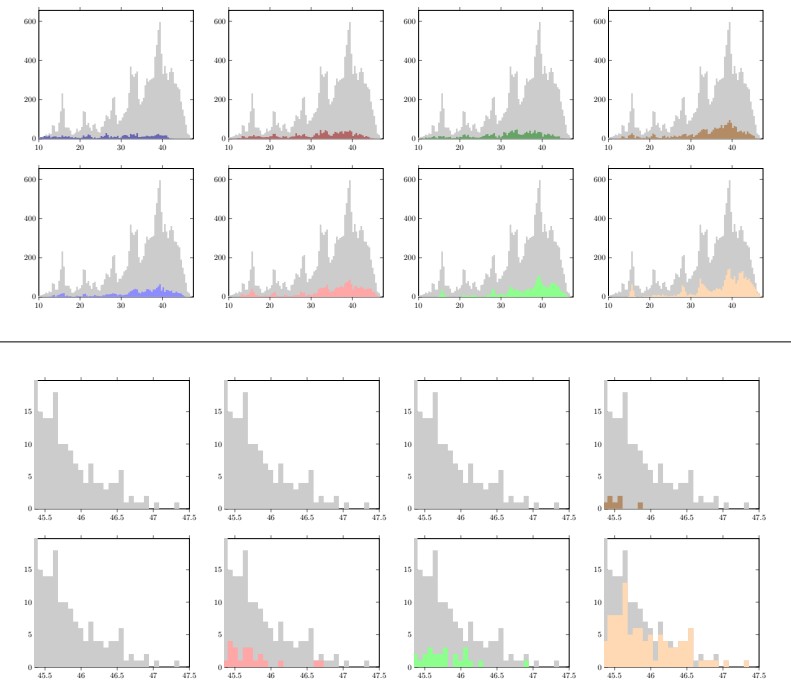

### D.3.5 `grasp`.

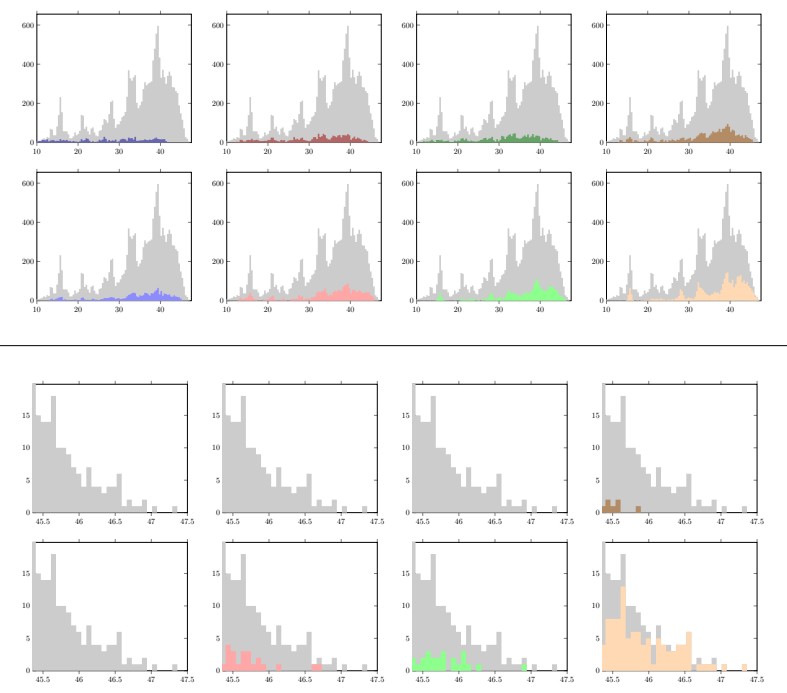

### D.3.6 `synflow`.

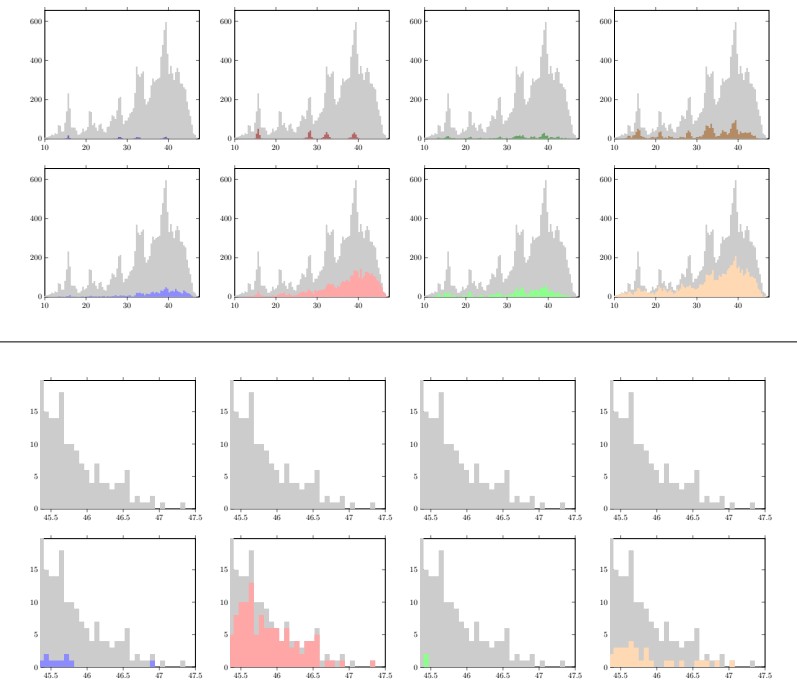

### D.3.7 `ntk`.

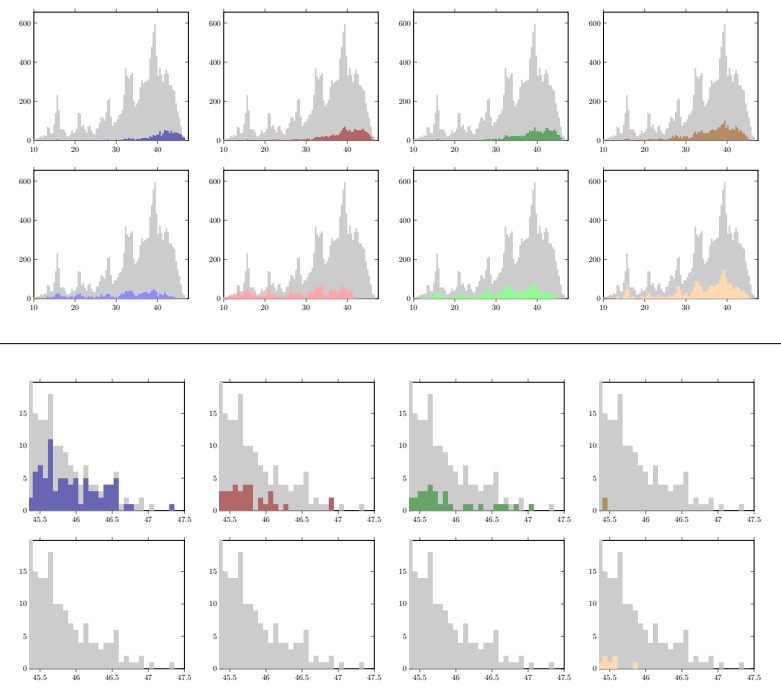

### D.3.8 `#lr`.

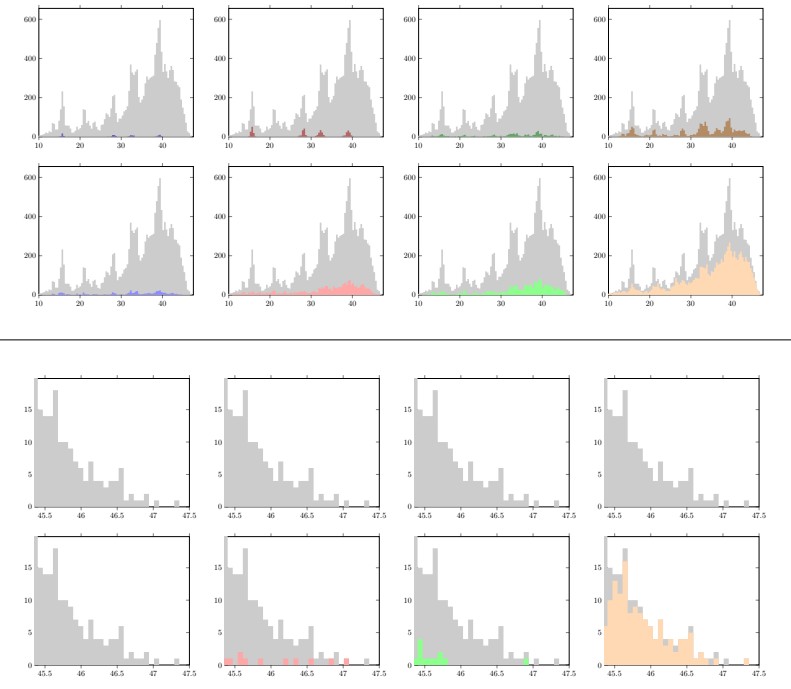

### D.3.9 #params.

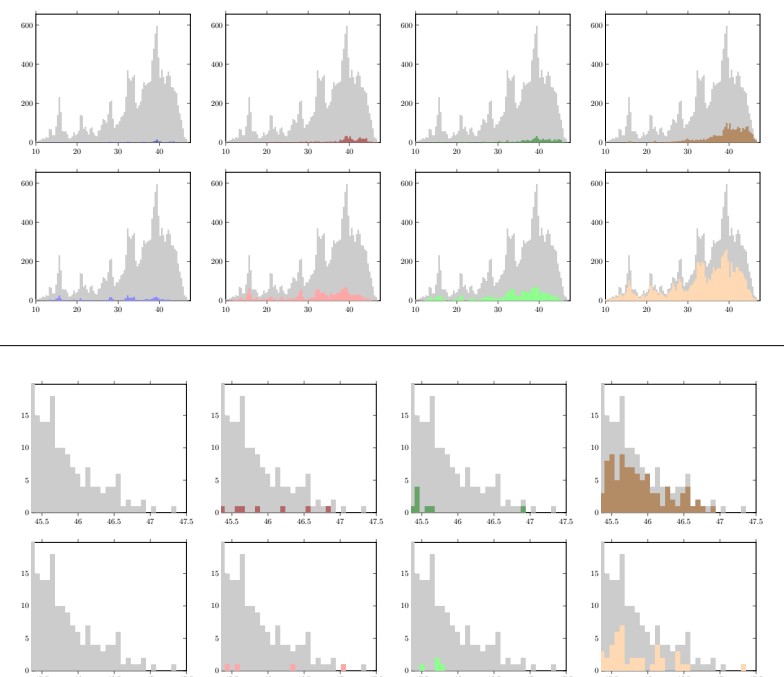