# OpenReview forum: "Analyzing Few-Shot Neural Architecture Search in a Metric-Driven Framework"
_automl.cc/AutoML/2024/Conference — AutoML 2024_

### Official Review · Reviewer_ZSkN · 2024-03-27

**Potential Impact On The Field Of Automl Rating:** 2
**Technical Quality And Correctness Rating:** 2
**Clarity Rating:** 2
**Ethics And Accessibility Rating:** Yes, regarding legal compliance (e.g.…

**Summary Of Contributions:**

The main contributions of the paper "Analyzing Few-Shot Neural Architecture Search in a Metric-Driven Framework" are as follows:

Introduction of Few-Shot NAS: The paper introduces the concept of Few-Shot Neural Architecture Search (NAS) as a method to reduce the costs associated with traditional NAS techniques by splitting the supernet into sub-supernets trained separately with reduced weight-sharing.

Generalization of Splitting Framework: The authors extend and generalize the Few-Shot NAS method by developing a framework that supports any arbitrary architecture evaluation metric. This framework allows for the generation of diverse splits, enhancing the efficiency of architecture evaluation in NAS.

Integration of Zero-Shot NAS Metrics: The paper incorporates various metrics from the zero-shot NAS literature into the splitting framework, providing insights into the benefits of splitting across different algorithms and metrics. Metrics such as jacobcov, gradnorm, snip, grasp, synflow, ntk, #lr, and #params are utilized to improve the performance of NAS algorithms.

Evaluation of Supernet Selection Techniques: The authors evaluate supernet selection methods and highlight the inadequacy of sub-supernet accuracy as a proxy for finding optimal sub-spaces. The study reveals that selecting sub-supernets based on validation accuracy may degrade the final performance of few-shot NAS.

Overall, the paper contributes to the advancement of NAS techniques by proposing a metric-driven framework for analyzing few-shot NAS methods, exploring the effectiveness of splitting strategies, and evaluating the impact of different metrics on architecture search performance.

**Actions Required To Increase Overall Recommendation:**

Robustness Analysis:

Conduct additional experiments to assess the robustness of the proposed framework across different datasets or experimental settings.
Include ablation studies or sensitivity analyses to evaluate the impact of different components of the approach on performance.
Comparison with State-of-the-Art:

Include a more detailed comparison with existing state-of-the-art methods in the field of AutoML to highlight the novelty and effectiveness of the proposed framework.
Discuss how the proposed approach improves upon or complements existing methods in the literature.

**Clarity:**

The paper "Analyzing Few-Shot Neural Architecture Search in a Metric-Driven Framework" presents its contributions with clarity, outlining the novel metric-driven framework for analyzing few-shot NAS methods and the integration of various metrics from the zero-shot NAS literature. The paper effectively communicates the importance of splitting the supernet into sub-supernets and the impact of different metrics on architecture search performance.

Suggestions for further improving clarity:

Define Terminology: Ensure that all technical terms and acronyms are clearly defined for readers who may not be familiar with the specific terminology used in the field of AutoML.

Structured Presentation: Consider organizing the paper in a structured manner, with clear subsections for the introduction, methodology, experiments, results, and conclusions to help readers navigate the content more easily.

Visual Aids: Incorporating visual aids such as diagrams, tables, or figures to illustrate key concepts, experimental setups, or results can enhance the clarity of the paper and make complex information more accessible to readers.

Plain Language: Strive for clear and concise language throughout the paper, avoiding overly technical jargon or complex sentences that may hinder understanding.

Conclusion Recap: Provide a concise recap of the main contributions and findings in the conclusion section to reinforce the key takeaways for readers.

By addressing these suggestions and ensuring a clear and structured presentation of the contributions, the paper can further enhance its clarity and readability for a wider audience in the field of AutoML.

**Overall Review:**

The paper "Analyzing Few-Shot Neural Architecture Search in a Metric-Driven Framework" presents a novel metric-driven framework for analyzing few-shot NAS methods, focusing on the splitting of the supernet into sub-supernets for improved architecture search. Here is an overall review highlighting the positive and negative aspects of the paper:

Positive Aspects:

Novel Framework: The paper introduces a novel metric-driven framework for analyzing few-shot NAS methods, which addresses the challenge of splitting the supernet into sub-supernets. This framework provides a structured approach to evaluating the impact of different metrics on architecture search performance.

Integration of Zero-Shot NAS Metrics: By integrating various metrics from the zero-shot NAS literature, the paper enriches the analysis and provides a comprehensive evaluation of the splitting strategies. This integration adds depth to the study and offers insights into the effectiveness of different metrics in the context of few-shot NAS.

Comprehensive Experiments: The paper conducts comprehensive experiments to evaluate the impact of splitting on the performance of one-shot NAS algorithms using different metrics. The comparison against a random baseline and the reporting of results across multiple datasets demonstrate a thorough experimental design.

Clear Presentation: The paper effectively communicates the contributions, methodology, experiments, and conclusions in a clear and structured manner. The logical flow of the paper aids in understanding the proposed framework and the experimental results.

Negative Aspects:

Lack of Code Availability: While the code used for generating the splits is provided, the absence of open-sourced implementations of the NAS algorithms used in the study limits the reproducibility of the experiments. Providing access to all code and data is essential for ensuring reproducibility and transparency in research.

Limited Robustness Analysis: The paper could benefit from a more detailed analysis of the robustness of the proposed framework and metrics across different datasets or experimental settings. A more extensive evaluation of the framework's generalizability would strengthen the validity of the findings.

Comparison with State-of-the-Art: While the paper introduces a novel framework and integrates zero-shot NAS metrics, a direct comparison with existing state-of-the-art methods in the field of AutoML could further highlight the contributions and novelty of the proposed approach.

In conclusion, the paper makes significant contributions to the field of AutoML through its novel metric-driven framework for analyzing few-shot NAS methods. Addressing the limitations related to code availability, robustness analysis, and comparison with existing methods could further enhance the impact and credibility of the research.

**Potential Impact On The Field Of Automl:**

The paper "Analyzing Few-Shot Neural Architecture Search in a Metric-Driven Framework" has the potential to significantly impact the field of Automated Machine Learning (AutoML) due to the following reasons:

Advancement of Few-Shot NAS Techniques: The introduction of Few-Shot NAS and the development of a metric-driven framework for analyzing architecture search methods contribute to enhancing the efficiency and effectiveness of AutoML algorithms. This novel approach to splitting the supernet into sub-supernets trained separately can lead to improved neural network architectures.

Generalization of Splitting Framework: By generalizing the splitting framework to accommodate various architecture evaluation metrics, the paper provides a versatile tool for researchers and practitioners in the AutoML field. This flexibility allows for the exploration of diverse splitting strategies and metrics, potentially leading to better-performing architectures.

Integration of Zero-Shot NAS Metrics: The incorporation of metrics from the zero-shot NAS literature expands the evaluation criteria available for architecture search. By leveraging these metrics in the splitting framework, the paper offers new insights into optimizing NAS algorithms and improving architecture search performance.

Evaluation of Supernet Selection Techniques: The critical evaluation of supernet selection methods and the identification of flaws in using sub-supernet accuracy as a proxy for sub-space selection highlight important considerations for researchers in the AutoML domain. This analysis can guide future research in developing more robust and effective strategies for selecting optimal architectures.

Given these contributions, it is likely that researchers and practitioners in the AutoML field will cite this paper for its insights into few-shot NAS techniques, the development of a flexible framework for architecture evaluation, and the evaluation of supernet selection methods.

**Review Confidence:**

4

**Review Rating:**

6

**Review Summary:**

The paper "Analyzing Few-Shot Neural Architecture Search in a Metric-Driven Framework" presents a novel metric-driven approach for analyzing few-shot NAS methods, focusing on supernet splitting and metric integration. The paper demonstrates clear contributions, comprehensive experiments, and a well-structured presentation. However, limitations in code availability, robustness analysis, and comparison with state-of-the-art methods were noted.

Recommendation: I recommend acceptance with minor revisions. The paper's contributions and experimental design are strong, but addressing the limitations related to code availability, robustness analysis, and comparison with existing methods would further enhance the research's impact and reproducibility. With these minor revisions, the paper can make a valuable contribution to the field of AutoML.

**Technical Quality And Correctness:**

The proposed approach, theory, experiments, and conclusions presented in the paper "Analyzing Few-Shot Neural Architecture Search in a Metric-Driven Framework" appear to be sound and of high quality. Here are some key points regarding the technical quality and correctness of the paper:

Approach and Theory: The paper introduces a novel metric-driven framework for analyzing few-shot NAS methods, which addresses the challenge of splitting the supernet into sub-supernets for improved architecture search. The integration of various metrics from the zero-shot NAS literature adds depth to the analysis and provides valuable insights into the effectiveness of different splitting strategies.

Experiments: The experiments conducted to evaluate the impact of splitting on the performance of one-shot NAS algorithms using different metrics are comprehensive and well-structured. The comparison against a random baseline and the reporting of results across multiple datasets demonstrate a thorough experimental design.

Conclusions: The conclusions drawn from the experimental results are supported by the data presented in the paper. The observation that splitting generally leads to better architecture discovery across various NAS algorithms is a significant finding that contributes to the understanding of few-shot NAS techniques.

Perceived Flaws: While the overall technical quality of the paper appears to be strong, there are a few potential areas for improvement or further exploration:

Robustness Analysis: It would be beneficial to include a more detailed analysis of the robustness of the proposed framework and metrics across different datasets or experimental settings to ensure the generalizability of the findings.

Comparison with State-of-the-Art: Providing a comparison of the proposed approach with existing state-of-the-art methods in the field of AutoML could further strengthen the paper's contributions and highlight its novelty.

Overall, the paper demonstrates a high level of technical quality and correctness in its approach, experiments, and conclusions, with opportunities for further exploration and validation in future research.

---

### Official Review · Reviewer_BudW · 2024-03-28

**Potential Impact On The Field Of Automl Rating:** 2
**Technical Quality And Correctness Rating:** 2
**Clarity Rating:** 2
**Actions Required To Increase Overall Recommendation:** 1. The writing should be improved fir…

**Summary Of Contributions:**

To resolve the unreasonable cost issue of typical NAS methods, this paper introduces a few-shot NAS setting. They verify several assumptions in the few-show / one-shot NAS setting: 1. supernet splitting is indeed beneficial; 2. splitting is unable to isolate the best architectures; 3. commonly used proxies to subspace selection are flawed and cannot recognize a good supernet.

**Clarity:**

The writing needs to be improved. Some expressions are confusing and miss importation explanations or definitions. For example:
1. In the abstract section, authors mention one-shot NAS, few-shot NAS, GM-NAS and zero-shot NAS. It is a little bit hard to follow the logical flow and understand the motivation and definition of the task.
2. Figure 1 (a) is also confusing. Why do you design such a splitting tree? What are the relationships between different nodes in the tree? What is the difference between your method and previous methods? The caption should be self-consistent without any knowledge prior.

**Overall Review:**

This paper does a good experimental analysis for few-shot NAS. However, the motivation and contributions should be addressed more clearly and the writing also needs to be improved. Existing questions are listed in the above.

**Potential Impact On The Field Of Automl:**

The existing challenge is interesting for me. This paper does an in-depth experimental analysis of few-shot NAS. However, it is not clear to explain the motivation of why authors propose such a method. What is the difference between your method and GM-NAS? Does your method focus on both few-shot NAS and zero-shot NAS? I raise many questions with the abstract section and have concerns about the potential impact.

**Reproducibility:**

They provide code for reproducibility.

**Review Confidence:**

4

**Review Rating:**

4

**Review Summary:**

This paper focuses on an interesting topic for NAS. However, the authors didn't present the motivation, contributions and details of the method clearly. I believe it would be improved after a major revision. Unfortunately, I have to give a score of weak reject at this time.

**Technical Quality And Correctness:**

Some details and information are missing in this paper. For example:
1. In Line 51, there is no clear definition of "important assumption" for the few-shot NAS. The assumption is highly relevant to the contributions of this paper. In the meanwhile, there is no theoretical evidence or detailed discussion for the assumption.
2. In Line 191, this paper addresses the difference between the proposed method and previous work GM-NAS, "we do not select an edge to split at run time, instead opting to always split the first three edges of the supernet in that order". This part is still unclear to me. What is the motivation behind this? Is it novel to improve GM-NAS with different metrics and splitting? The authors should address their own contributions with clear motivation.

---

### Official Review · Reviewer_5LnU · 2024-03-28

**Potential Impact On The Field Of Automl Rating:** 3
**Technical Quality And Correctness Rating:** 4
**Clarity Rating:** 4

**Summary Of Contributions:**

The paper delves into the realm of few-shot Neural Architecture Search (NAS) methods, focusing specifically on the technique of splitting the one-shot NAS supernetwork into sub-supernetworks. This subdivision aims to simplify the search for optimal architectures, particularly facilitating gradient-based NAS optimization. By evaluating the methodologies proposed in two seminal works [1] and [2], the paper scrutinizes the efficacy of sub-supernetwork splits in mitigating co-adaptation during the search process. Additionally, it explores the utilization of zero-cost proxy metrics, shedding light on their potential in the few-shot NAS paradigm. Notably, the study unearths empirical evidence suggesting that the success of few-shot NAS methods may not solely stem from co-adaptation alleviation via sub-supernetwork splits, but rather from the reduced search space itself. Furthermore, it uncovers shortcomings in the evaluation phase of certain sub-supernetwork techniques, thereby enriching the understanding of the intricacies involved in few-shot NAS methodologies.

**-- References --**

[1] Y. Zhao, L. Wang, Y. Tian, R. Fonseca, and T. Guo. Few-shot neural architecture search. In ICML 2021.

[2] S. Hu, R. Wang, L. Hong, Z. Li, C.-J. Hsieh, and J. Feng. Generalizing few-shot NAS with gradient matching. In ICLR 2022.

**Actions Required To Increase Overall Recommendation:**

- Run the same evaluation on another search space (e.g. another tabular NAS benchmark)

- Run the evaluation in Section 4.3 with the other metrics used in the paper, i.e. grasp, ntk, etc.

**Clarity:**

The paper is well-written and I enjoyed reading it. There are no major issues in the writing and structure.

-- Minor --

In lines 248 - 251, the authors mention the distribution of the other evaluation metrics. Would be great to have a figure similar to Fig. 3 in the main paper, where every plot corresponds to one metric and shows the ECDFs (cumulative distributions) of every sub-supernet split with different colors.

**Overall Review:**

This paper shows some interesting findings regarding few-shot NAS approaches. Below I list some of the positive and negative points of the paper:

Pros:

1. **Clear Motivation and Objectives:** The paper effectively articulates the rationale behind investigating few-shot NAS methods and delineates clear research objectives. This clarity enhances the reader's understanding and underscores the significance of the study.

2. **Insightful Results and Analysis:** The findings presented in the paper offer valuable insights into the underlying mechanisms of few-shot NAS approaches. By dissecting the efficacy of sub-supernetwork splits and zero-cost proxy metrics, the study contributes to a nuanced understanding of these methodologies.

3. **Robust Experimental Protocol:** The experimental setup is well-designed and executed, bolstering the reliability and reproducibility of the results.

4. **First to evaluate Zero-cost Proxies in few-shot NAS:** The evaluation of zero-cost proxy metrics within the context of few-shot NAS is new. This novel exploration expands the methodological toolkit available for assessing NAS methodologies and opens avenues for further research in this domain.

5. **Clarity of Presentation:** The writing style employed in the paper is lucid and accessible, ensuring that complex concepts are conveyed effectively to the reader. This clarity enhances the readability and comprehensibility of the manuscript.

6. **Availability of Code:** The provision of code accompanying the paper fosters transparency and facilitates the replication of experiments by other researchers. This openness enhances the reproducibility of the study's findings.

Cons:

1. **Limited Scope of Search Spaces:** A notable limitation of the study is the restricted scope of search spaces examined. While the experiments provide valuable insights, extending the analysis to encompass diverse NAS benchmarks such as the ones in NAS-Bench-Suite (https://arxiv.org/abs/2201.13396) or NAS-Bench-Suite-Zero (https://arxiv.org/abs/2210.03230) would bolster the generalizability and robustness of the conclusions. This way, their claims such as "while some metrics may separate architectures in more advantageous ways, the performance gain could simply be attributed to the search taking place in a smaller search space, rather than inherent co-adaptation reduction inside the sub-supernets", will be better suported.

**Potential Impact On The Field Of Automl:**

I think the experimental results of this paper are definitely interesting and raise valid question on the effectiveness and claims of the few-shot NAS approaches. The findings presented in this paper hold implications for the advancement of few-shot Neural Architecture Search (NAS) methodologies and their application in practical settings. By scrutinizing the efficacy of sub-supernetwork splits and zero-cost proxy metrics, this study provides valuable insights that can inform the design and optimization of neural network architectures with limited computational resources. The clear articulation of research objectives, coupled with sound experimental protocols and accessible presentation, positions this work as a valuable paper in the understanding of few-shot NAS techniques. Addressing the limitation regarding the scope of search spaces used in the paper would further amplify the impact of this research, enabling more comprehensive evaluations and fostering greater confidence in the generalizability of the findings.

**Reproducibility:**

+ The authors provide the code in the supplementary material. I do not see any potential issues arising regarding reproducibility.

**Review Confidence:**

5

**Review Rating:**

7

**Review Summary:**

Overall, I think this paper provides a useful empirical study of few-shot NAS methods, with results demonstrating that some of the claims from previous work might not be correct. It also investigates the applicability of zero-cost proxies inside the few-shot NAS realm. The clear articulation of research objectives, insightful findings, experimental protocol, and accessibility, render it a valuable addition to the existing literature. I lean towards acceptance, even though the authors need to provide another search space where their empirical findings are the same.

**Technical Quality And Correctness:**

The motivation to conduct the experiments in the paper is clear and meaningful. The authors utilized the NB201 benchmark in order to verify their claims and draw conclusions only on that. While this is a very useful benchmark, there are also other NAS tabular benchmarks that the authors could have conducted their experimental evaluation. See NAS-Bench-Suite [3] for a list. The findings are interesting, however, I would like to see another benchmark where these claims hold and the results are consistent with the ones in NB201.

---

### Official Review · Reviewer_x6rh · 2024-03-31

**Potential Impact On The Field Of Automl Rating:** 4
**Technical Quality And Correctness Rating:** 2
**Clarity Rating:** 4

**Summary Of Contributions:**

- Verifying that splitting the supernet into sub-spaces does help one-shot NAS algorithms find better-performing architectures compared to searching the full supernet space.
- Assessing the impact of using different metrics/objectives in the splitting procedure.
- Observing that despite splitting, the sub-spaces created are unable to isolate/contain the globally best architectures separately, which limits the synergies with search algorithms.
- Showing that commonly used proxies/heuristics for selecting promising sub-spaces, like the accuracy of the supernet itself, are flawed and cannot reliably identify good supernets or indicate which ones contain the best architectures.

**Actions Required To Increase Overall Recommendation:**

- Provide an ablation study for the phenomenon observed by using accuracy as a selection criterion. For example, when authors argue that strict splitting conditions are not necessary for GM-NAS, they must provide an in-depth theoretical explanation, cite a study, or prove with an ablation study why they think this way. Ablation studies must be conducted for such claims to prove the scientific efficacy of the interpretation for it to benefit the broader community.
- The conclusion needs a reframing to match the Abstract.

**Clarity:**

The problem is well defined and enough definitions are provided making it easy to follow the the paper. Experiments are appropriately elaborated with charts and tables appropriately labelled

**Overall Review:**

One-short and Few-shot NAS are significant areas of research for AutoML. This study focuses on analyzing the impact of using validation accuracy for subnet selection as well as investigating the benefits of splitting across 17 algorithms and metrics concluding supernet selection methods are flawed. However, the study does not provide any ablation study(if conducted) and merely claims based on tunning epochs. It's important to provide a theoretical explanation as to why authors observe this phenomenon to further prove the efficacy of their experimental design. If the output of their insight is based on building a novel framework that can generalize based on GM-NAS and other existing techniques, authors must describe this framework in detail and their unique contributions.

**Potential Impact On The Field Of Automl:**

These results highlight fundamental limitations in the few-shot neural architecture search paradigm that leverages supernet splitting. The inability to reliably isolate top architectures in distinct subspaces and the ineffectiveness of common supernet selection proxies pose challenges. This calls for a re-evaluation of the assumptions and heuristics used in few-shot NAS methods. The findings underscore the need for developing more principled techniques to generate promising subspaces and identify fruitful supernets. Addressing these issues could potentially unlock better performance and efficiency gains in AutoML systems that rely on few-shot NAS, shaping the future direction of research in this area.

**Review Confidence:**

3

**Review Rating:**

7

**Review Summary:**

One-short and Few-shot NAS are significant areas of research for AutoML. This study focuses on analyzing the impact of using validation accuracy for subnet selection as well as investigating the benefits of splitting across several algorithms and metrics concluding supernet selection methods are flawed. However, the study does not provide any ablation study(if conducted) and merely claims based on tunning epochs. It's important to provide a theoretical explanation as to why authors observe this phenomenon to further prove the efficacy of their experimental design. If the output of their insight is based on building a novel framework that can be generalized based on GM-NAS, authors must describe this framework in detail, highlight their unique contributions and how some of the changes made in Section 3.2 affect the outcome. However, the insights mentioned in the paper are very interesting and could benefit not only AutoML group of researchers but non-AutoML researchers as well.

**Technical Quality And Correctness:**

The study is performed on multiple datasets(CIFAR10, CIFAR100, IMAGENET16-120. This study claims that the architectures are not advantageous in the most optimal way and the proposed supernet selection methods are flawed. The study considered test accuracy for the datasets however, it was unclear how the accuracy metric is not a good proxy for the selection method on a validation set, The study does not provide enough theoretical explanation and/or ablation study to understand the cause and the effect.

---

### Meta-Review · Area_Chair_jYrv · 2024-04-22

**Paper Recommendation:** Accept
**Confidence:** 4

**Metareview:**

The authors proposed an extension of GM-NAS to be able to leverage arbitrary architecture evaluation metric. The method is clearly stated and evaluated with a diverse variety of metrics. The method is well motivated and looks reasonable. One critical question is about the practicality of the proposal. Looking at the results, the gradient matching in GM-NAS is already robust enough from the table. Then in practical, why not just using GM-NAS? If we choose this new approach, what is the best practice to efficiently secure a reliable and robust result? It would be great for the authors to discuss these in depth. Despite the practicality issue, considering the method contribution and analysis, I would recommend "accept".

General Suggestions:
- Reproducibility reviewer has difficulty to fully execute (some work and some don't). It would be great to continue improve the code quality and documentation.
- Ablation studies are critical to the paper, it would be great to move it to the main part and further enhance it.
- As an extension to GM-NAS, it is great to make it more general, while practicality concern is not addressed. If it can not be addressed in this paper, it would be great to have a in-depth discussion for this aspect.

---

### Decision · Program_Chairs · 2024-04-29

**Decision:**

Accept

**Comment:**

Thank you for submitting your paper. We are happy to tell you that we accept your paper to the main track. See you in Paris.